# ChemOrch: Empowering LLMs with Chemical Intelligence via Synthetic Instructions

Yue Huang[1],[*] Zhengzhe Jiang[*], Xiaonan Luo[1], Kehan Guo[1], Haomin Zhuang[1], Yujun Zhou[1]
Zhengqing Yuan[1], Xiaoqi Sun[2], Jules Schleinitz[3], Yanbo Wang[4], Shuhao Zhang[5]
Mihir Surve[6], Nitesh V Chawla[1], Olaf Wiest[6], Xiangliang Zhang[1]
[1]Department of Computer Science and Engineering, University of Notre Dame
[2]MIT   [3]CalTech   [4]MBZUAI   [5]CMU
[6]Department of Chemistry & Biochemistry, University of Notre Dame

## Abstract

Empowering large language models (LLMs) with chemical intelligence remains a challenge due to the scarcity of high-quality, domain-specific instruction-response datasets and the misalignment of existing synthetic data generation pipelines with the inherently hierarchical and rule-governed structure of chemical information. To address this, we propose **ChemOrch**, a framework that synthesizes chemically grounded instruction–response pairs through a two-stage process: task-controlled instruction generation and tool-aware response construction. ChemOrch enables controllable diversity and levels of difficulty for the generated tasks, and ensures response precision through tool planning & distillation, and tool-based self-repair mechanisms. The effectiveness of ChemOrch is evaluated based on: 1) the **high quality** of generated instruction data, demonstrating superior diversity and strong alignment with chemical constraints; 2) the **reliable generation of evaluation tasks** that more effectively reveal LLM weaknesses in chemistry; and 3) the significant **improvement of LLM chemistry capabilities** when the generated instruction data are used for fine-tuning. Our work thus represents a critical step toward scalable and verifiable chemical intelligence in LLMs. The code is available at https://github.com/HowieHwong/ChemOrch.

## 1 Introduction

Large Language Models (LLMs) have exhibited exceptional capabilities across a wide range of tasks, to be widely applied in various downstream tasks [1–5]. Among these, chemistry represents a particularly promising field where LLMs can assist in accelerating molecular design [6, 7], facilitating scientific discovery [8, 9], and democratizing access to expert-level chemical knowledge [10, 11]. Empowering LLMs with strong chemical reasoning capabilities could significantly impact areas such as drug discovery, materials development, and organic synthesis [12, 13].

Despite this potential and existing efforts [9, 10], empowering LLMs with chemistry domain knowledge remains challenging. First of all, there exists Challenge 1: **data scarcity in both training and testing**, as disclosed in recent benchmarking studies [14]: LLMs have not yet achieved the level of performance expected by chemistry scientists. High-quality instruction datasets for guiding LLMs to learn chemistry knowledge are extremely limited. While chemistry-related corpora have been utilized during pretraining [4], constructing task-specific fine-tuning datasets and fine-grained evaluation sets often requires intensive expert annotation [15], leading to high costs and limited scalability.

To overcome data scarcity, synthetic data generation offers a promising alternative. However, there exists Challenge 2: **mismatch between general-purpose synthetic frameworks and chemistry-**

---

[*]Equal Contribution

**specific requirements.** Most existing instruction generation frameworks are built for general text understanding, and are fundamentally misaligned with the structured and rule-bound nature of chemical tasks [16–18]. Chemical problems often involve operations over molecular structures, require strict adherence to conservation laws or valence constraints, and demand accurate function grounding. Even minor errors, e.g., invalid atoms or incorrect stereochemistry, can lead to chemically meaningless or unreliable results, a failure mode less critical in standard NLP.

Even when domain-specific generation is attempted, there remains Challenge 3 for **ensuring diversity, executability, and controllability in the data synthesis process**. Effective chemical instruction datasets must span a wide range of tasks, from basic property prediction to complex retrosynthesis planning, while ensuring that generated responses are chemically valid and verifiable. Beyond diversity, maintaining *executability*, the ability for model outputs to conform to domain rules and withstand external verification, is crucial for scientific reliability. Furthermore, controlling the difficulty, specificity, and complexity of generated instructions is important for effective training of LLMs (enabling progressive skill development and robust understanding), but remains particularly difficult in current automated instruction generation pipelines.

To address these challenges, we propose **ChemOrch**, a framework for constructing synthetic instruction-response pairs that enable LLMs to acquire chemistry domain knowledge. ChemOrch consists of a two-stage pipeline: (1) **Task-Controlled Instruction Generation**. Given a chemistry task (e.g., property prediction) along with user-defined constraints (e.g., target difficulty, required keywords) and metadata (e.g., reference files, extra tool configuration), ChemOrch enables the generation of diverse and controllable instructions (e.g., different molecules with various types of properties to predict, at varying levels of prediction difficulty). While the diversity is ensured by specifying constraints, the difficulty level is controlled by a *difficulty reward model with feedback*, which evaluates and iteratively refines instructions to align with user-specified complexity levels. (2) **Tool-Aware Response Construction**. To ensure precise responses, ChemOrch leverages a set of tools to ground its outputs, as certain chemical tasks are straightforward for these tools but challenging for LLMs due to their lack of domain knowledge. For example, tasks like name translation can be easily handled by chemical tools, but are difficult for LLMs to generate accurately. For the given instruction generated before, ChemOrch decomposes the associated problem into intermediate reasoning steps, retrieves and distills relevant tools, and generates code scripts to produce accurate outputs. This process includes multi-stage self-repair mechanisms and sufficiency checks, ensuring that generated responses are verifiable, executable, and faithfully satisfy the original instruction intent (e.g., see several instruction-response examples in Figure 15, Figure 16, Figure 17).

We conduct extensive experiments based on **ChemOrch**, evaluating its effectiveness across multiple dimensions, including the diversity, response quality, and constraint adherence of the generated instruction-response pairs, and its usefulness on two important applications. First, ChemOrch serves as a reliable evaluation framework that identifies LLM weaknesses, and enables scalable, task-specific assessments with high fidelity. Second, it enhances LLM performance in chemical QA and reasoning tasks, when the generated instructions are used for fine-tuning.

Overall, our contributions are threefold: **1)** We introduce ChemOrch, breaking the limits of instruction data scarcity to enable LLMs to solve chemistry-related challenges more effectively. The synthetic instruction-response pairs are diverse in topics, challenging at controllable levels, and comprehensive for covering a wide range of chemical tasks with precise answers guaranteed, as validated by human experts. **2)** We propose a novel synthesis framework featuring a two-stage pipeline, incorporating difficulty control, tool decomposition, and distillation, as well as self-repair mechanisms, allowing scalable and high-quality instruction response generation. **3)** Importantly, we showcase the significant impact of ChemOrch on two key applications: facilitating chemistry evaluation and improving the chemistry capability of LLMs, demonstrating the effectiveness of our framework.

## 2 Preliminary: Harnessing Chemical Tools within ChemOrch

Leveraging chemistry tools is a core aspect of our work, ensuring that the responses in the curated instruction dataset are both accurate and reliable. Furthermore, the tool execution process itself can be utilized to evaluate and enhance LLM's proficiency in using tools—an important long-term objective of our research (Our experiments presented in section 4 demonstrate that ChemOrch significantly improves LLMs' tool usage capabilities). Before introducing the framework of ChemOrch, this section outlines how chemical tools are constructed and integrated.

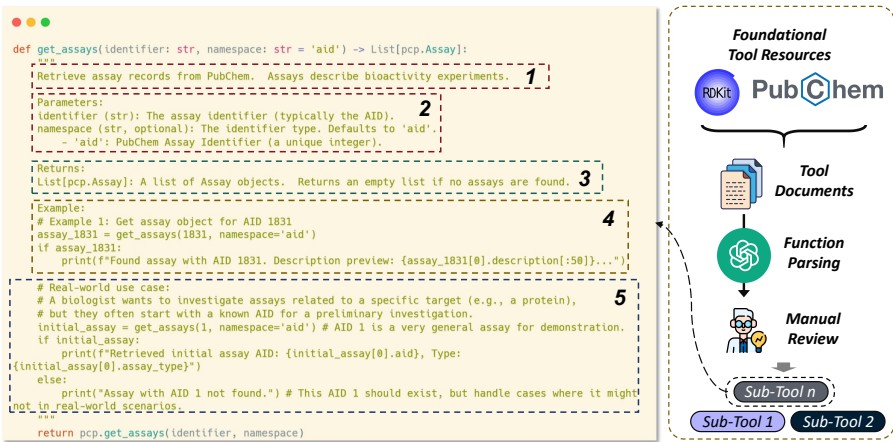

Figure 1: An example of a sub-tool (left) and the tool decomposition pipeline (right). Each sub-tool includes five components: 1) *a high-level operation description*, 2) *formal argument specification*, 3) *expected return values*, 4) *minimal working example*, and 5) *real-world use case*.

**Tools Overview.** ChemOrch leverages two categories of tools: chemistry-related tools such as RDKit [19] and PubChem [20] and general-purpose tools that include web search and the Python environment for code scripting. More details about tools selection are provided in Appendix I.

**Motivation.** In real-world scenarios, humans typically interact with chemistry tools at the function level, selecting specific operations rather than engaging with the tool as a monolithic system. Inspired by this, we aim to enhance the usability of foundational chemistry tools (e.g., RDKit) in instruction-driven response generation by decomposing them into fine-grained sub-tools. This decomposition serves three purposes: (1) it enables precise control by isolating individual functions, (2) it simplifies tool semantics, making them more interpretable and accessible for LLMs, and (3) it mitigates the challenges LLMs face when handling complex, multi-step operations with minimal context.

**Sub-Tool Construction.** We begin by feeding the official documentation of RDKit and PubChem into an LLM (i.e., GPT-4o) and applying a few-shot learning approach to guide the extraction of function-level operations. These operations, which represent the atomic units of functionality, are identified based on their relevance to common chemistry tasks. The LLM is then prompted to synthesize these operations into callable Python code, each encapsulated as a **sub-tool**. Every sub-tool is constructed following a schema consisting of five components, as illustrated in Figure 1. This semi-automated process significantly accelerates tool decomposition while maintaining structural consistency. To ensure correctness and usability, we subsequently perform a manual review of all generated sub-tools, with review procedures introduced in Appendix C. In total, we constructed **74 sub-tools, 57 derived from RDKit** and **17 from PubChem**. For each sub-tool, we provide both a minimal working example and a real-world use case. The real-world use case demonstrates how the operation is applied in meaningful task contexts, e.g., using a *molecular descriptor computation sub-tool* within a *solubility prediction pipeline*. The examples help LLMs understand how to invoke a sub-tool and why and when to use it. These sub-tools are integrated with other general-purpose tools to form a **tool pool**, which is then used in the instruction-response generation process of ChemOrch.

**Extensibility.** While ChemOrch currently integrates a limited set of tool types, it is designed to be extensible. A key enabler of this extensibility is the use of *metadata* (as shown in Figure 2), which allows users to specify additional tool configurations at runtime. For instance, users can upload custom wrappers or specify endpoints for private APIs, as shown in Appendix N. By doing this, ChemOrch is able to dynamically recognize and utilize new capabilities, enabling broader coverage of specialized chemistry tasks.

## 3 ChemOrch

We formalize ChemOrch as a framework consisting of a two-stage generation pipeline that produces instruction–response pairs tailored for chemistry-related tasks.

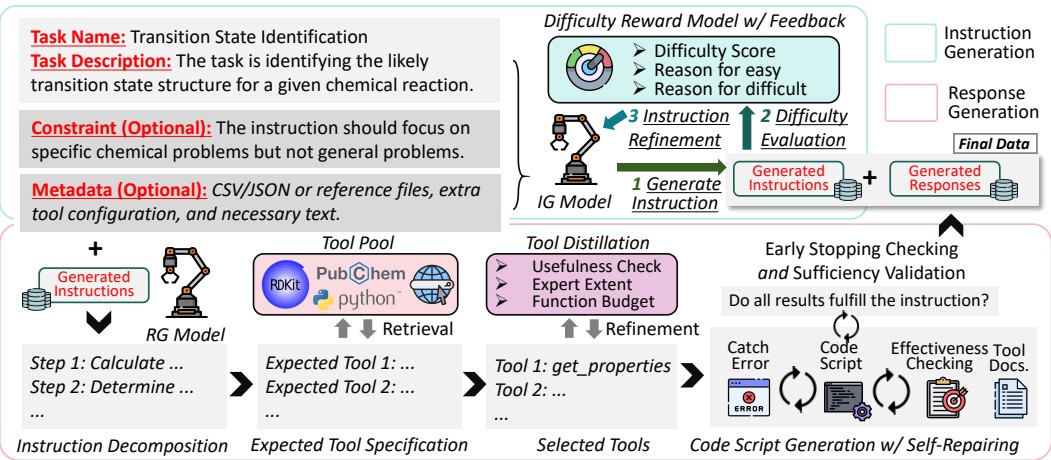

Figure 2: ChemOrch Framework. It consists of instruction generation (IG) with difficulty evaluation, and response generation (RG) by calling tools from the tool pool.

## 3.1 Instruction Synthesis

Let $\mathcal{T}$ denote the space of chemistry-related tasks, and let $\mathcal{C}$ represent a set of user-defined constraints (e.g., instruction length, complexity, keyword requirements, formatting, or few-shot examples for guiding generation) that enable more controllable generation. In addition to constraints, we also introduce a metadata set $M$, which includes auxiliary inputs such as reference files, specialized tool configurations, or essential textual descriptors. These metadata items serve as seed data to guide and condition the instruction generation process more precisely (e.g., by providing molecular structures as input when generating instructions for reaction prediction tasks), as shown in top-left of Figure 2.

Given a task $t \in \mathcal{T}$, constraint $c \in \mathcal{C}$, and metadata $m \in M$, the instruction generation (IG) model $\mathcal{M}_{\text{inst}}$ (i.e., an LLM) synthesizes an instruction $x = \mathcal{M}_{\text{inst}}(t, c, m)$. The details of instruction generation are presented in Algorithm 1 in Appendix M. For all tasks in $\mathcal{T}$, by varying the constraint in each running iteration alongside the metadata (check implementation details in Appendix B), the IG model generates diverse instructions $\mathcal{X} = \{x_i\}_{i=1}^N$.

**Difficulty Controlling.** Previous studies have shown that controlling the complexity or difficulty of LLM-generated instructions remains a non-trivial challenge [21, 22]. Maintaining calibrated difficulty levels in the chemistry domain is essential for supporting progressive skill acquisition.

To address this, we introduce a *difficulty reward model with feedback*, denoted $\mathcal{M}_{\text{diff}}$, which evaluates the generated instruction $x$ and outputs both a scalar difficulty score $d$ (from 1–5) and a set of localized difficulty explanations: $(d, e) = \mathcal{M}_{\text{diff}}(x)$, where $e$ indicates which aspects of the instruction (e.g., terminology, structure, domain scope) contribute to its simplicity or complexity.

This feedback is used to guide the instruction generation process (see Algorithm 1). Specifically, $\mathcal{M}_{\text{inst}}$ receives $(t, c, m, e)$ as input and is prompted to revise or regenerate $x$ such that the resulting difficulty better aligns with user intent or target distributions. We build $\mathcal{M}_{\text{diff}}$ by applying supervised fine-tuning (SFT) on Meta-Llama-3.1-8B-Instruct on 3,390 annotated samples. Evaluation by human experts indicates that $\mathcal{M}_{\text{diff}}$ successfully captures the nuances of chemistry instruction complexity. See more details of implementation and evaluation about $\mathcal{M}_{\text{diff}}$ in Appendix D.

## 3.2 Response Construction: Overall Procedure

Given an instruction $x$, the response generation (RG) model $\mathcal{M}_{\text{resp}}$ (i.e., an LLM) is responsible for generating an executable and chemically accurate output $y \in \mathcal{Y}$ for the associated question in $x$. Several examples of instruction-response pairs are presented in Figure 15 and Figure 16. To ensure factual grounding and modularity, $\mathcal{M}_{\text{resp}}$ does not directly generate free-form responses; instead, it leverages a tool pool $\mathcal{F} = \{f_1, f_2, \ldots, f_K\}$, as well as metadata $m$ (the same as used in the instruction generation), to generate the responses.

**Instruction Decomposition and Tool Planning.** The response model first decomposes the instruction into a sequence of intermediate reasoning steps, denoted as $s = \text{Decompose}(x) =$

$(s_1, s_2, \ldots, s_L)$. Each step $s_\ell$ reflects a sub-goal necessary to fulfill the instruction, such as data retrieval, property computation, or output formatting. Not all steps necessarily require tool usage.

The $\mathcal{M}_{\text{resp}}$ model considers the entire sequence of reasoning steps and generates a set of expected tool descriptions $\{d_1, d_2, \ldots, d_M\} \subset \mathcal{D}$, where each $d_m$ represents a distinct functional capability that may be needed to support one or more steps, denoted as $\{d_1, \ldots, d_M\} = \mathcal{M}_{\text{resp}}(s, m)$.

**Tool Retrieval and Distillation.** To map these expected tool descriptions to actual executable tools, semantic retrieval is performed over the tool pool. For each $d_m$, cosine similarity is computed between its embedding and the embeddings of available tools: $\text{Top-}k(d_m) = \underset{f_k \in \mathcal{F}}{\arg\,\text{top-}k} \; \cos(\phi(d_m), \phi(f_k))$.

The retrieval yields a candidate set $\mathcal{F}_m^{\text{raw}}$ for each expected tool description. The union of all retrieved candidates forms the raw tool pool for the response: $\mathcal{F}^{\text{raw}} = \bigcup_{m=1}^{M} \mathcal{F}_m^{\text{raw}}$.

Tool distillation is then performed globally using $\mathcal{M}_{\text{resp}}$, given the instruction $x$, reasoning steps $s$, metadata $m$, and candidate tools $\mathcal{F}^{\text{raw}}$. The model iteratively refines the toolset by eliminating redundant or ineffective tools, adhering to:

- **Usefulness Check**: Remove tools that do not contribute actionable results for subgoals.
- **Expert Extent**: Prefer tools that align closely with reasoning intents, minimizing auxiliary steps.
- **Tool Budget**: Enforce a size limit $\tau$ by pruning low-utility tools.

Formally denoted as: $\mathcal{F}^\star = \text{Distill}(\mathcal{F}^{\text{raw}}, x, s, m, \tau)$.

**Tool Execution and Answer Assembly.** With the distilled tool set $\mathcal{F}^\star$, the response model generates and executes scripts for each $f \in \mathcal{F}^\star$, incorporating metadata $m$ where applicable (e.g., tool configuration or guidance, molecule information). Outputs are denoted $o_f = f(a_f)$. The model then synthesizes the final response using the collected outputs: $y = \mathcal{M}_{\text{resp}}(x, \{o_f\}_{f \in \mathcal{F}^\star}, m)$.

We describe the detailed procedure of tool calling and validation in the following subsection.

### 3.3 Response Construction: Tool Calling

**Code Script Generation.** For each selected tool $f \in \mathcal{F}^\star$, the model generates an executable code script $\mathcal{S}_f$ based on tool specification, instruction, reasoning steps, and metadata, denoted as $\mathcal{S}_f = \mathcal{M}_{\text{resp}}(f, x, s, m)$, where metadata $m$ may contain pre-specified input files, configuration parameters, or external resources relevant to the tool.

**Self-Repairing.** Script execution errors are common due to issues such as incorrect input formats or API usage. We adopt a multi-stage self-repair protocol:

❶ Error Catching: If execution of $\mathcal{S}_f$ fails, the model captures the error trace $e$ and attempts repair:

$$\mathcal{S}_f^{(i+1)} = \mathcal{M}_{\text{resp}}(e, \mathcal{S}_f^{(i)}, m).$$

This process is repeated until success or a retry limit $R_{\max}$ is reached.

If all attempts fail, the model consults external documentation (e.g., from RDKit or PubChem) through web retrieval and regenerates the script based on the retrieved content and metadata.

❷ Effectiveness Checking: Even successful executions are not assumed to be sufficient. If the returned result does not meet user intent, e.g., missing keyword-level constraints or incorrect computation granularity, the model re-evaluates and refines the tool usage.

**Early Stopping & Sufficiency Validation.** After each tool execution, the model assesses whether the currently accumulated outputs $\{o_f\}$ already satisfy the instruction $x$. If so, the pipeline stops early, skipping the execution of remaining tools to improve efficiency.

After the finish of all selected tools, $\mathcal{M}_{\text{resp}}$ will check whether the outputs $\{o_f\}$ are incomplete or insufficient for fully answering the instruction. If yes, $\mathcal{M}_{\text{resp}}$ triggers a web-based retrieval step: $o_{\text{extra}} = \text{WebSearch}(x, m)$. The final output set used for response synthesis is:

$$\mathcal{O} = \{o_f\}_{f \in \mathcal{F}^\star} \cup \{o_{\text{extra}}\}.$$

The final response is constructed by composing all outputs in $\mathcal{O}$, grounded in the reasoning trace and enriched with any metadata-derived context. The details of response generation are shown in Algorithm 2 in Appendix M.

## 4 Experiments

### 4.1 Experiment Setup

**Models.** We employ GPT-4o [23] as the IG model across all experiments. For response generation, we adopt a hybrid setting: GPT-4o is used for general-purpose reasoning tasks such as decomposition, validation, and web retrieval. For components requiring fine-grained decision-making or complex reasoning (specifically, tool distillation, code script generation, and self-repairing), we utilize the o1-mini model [24], which demonstrates stronger reasoning capabilities. For text embedding, we adopt the text-embedding-3-small model [25]. All generation temperatures in our experiments are set to 1.0. Notably, ChemOrch is compatible with other models as IG/RG models as well; however, we adopt this configuration in our experiments to ensure consistency and facilitate controlled evaluation.

**Chemical Tasks in Instruction Generation.** We focus on two main categories of tasks: *General Chemistry Q&A* and *Task-Specific Challenges*. The former involves answering open-ended questions within the chemistry domain (e.g., *What are the main steps involved in synthesizing aspirin in the laboratory?*), which could be later used for evaluating/enhancing LLMs' chemistry knowledge and generation quality. The latter focuses on domain-specific tasks commonly used in existing benchmarks, requiring more sophisticated chemical reasoning and planning capabilities. These include property prediction (PP), molecule captioning (MC), name prediction (NP), and reaction prediction (RP), among others. See task examples in Table 11 and Table 12. Moreover, we introduce an agent-oriented task (tool usage) to assess LLM's capability on operating chemistry-related tools.

Due to space limitations, we refer the reader to Appendix B for additional details on experimental setup, including evaluation protocols, datasets used for analysis, training, and testing configurations.

### 4.2 Statistical Analysis of Generated Instruction–Response Pairs

**Word Count.** The distribution of generated data is shown in Figure 3. On average, each instruction contains 17.52 words, while the corresponding response is substantially longer, averaging 320.66 words, reflecting the detailed and comprehensive nature of model-generated outputs. Furthermore, each response utilizes approximately 1.24 tools on average (see examples in Figure 15, Figure 16).

**Topic Diversity and Coverage.** To assess the instruction dataset diversity and coverage, we have referred to the study [26, 27], and adopted two quantitative metrics: Average Pairwise Sample Similarity (APS) and Remote-Clique Score. APS captures the average similarity between sample instruction pairs–lower values indicate greater internal diversity. Remote-Clique identifies a maximally dissimilar subset of instruction samples, with higher scores suggesting broader input space coverage.

| Dataset | APS ↓ | Remote-Clique ↑ |
|---|---|---|
| ChemOrch (our) | 0.779 | 0.661 |
| ChemLLMBench [14] | 0.884 | 0.453 |
| Mol-Instructions [15] | 0.765 | 0.683 |
| ChemBench [28] | 0.784 | 0.613 |

Table 1: Diversity and coverage analysis, ChemOrch vs other datasets.

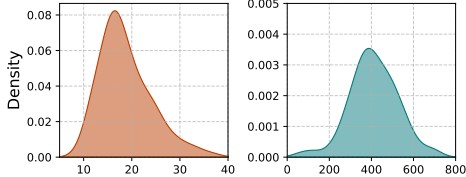

Figure 3: Word-count distribution of instructions (left) and responses (right).

As shown in Table 1, datasets generated by our proposed ChemOrch achieve significantly lower APS and higher Remote-Clique scores compared to ChemLLMBench [14] and ChemBench [28], suggesting that ChemOrch offers improved diversity and better structural spread in generation. Its diversity profile is comparable to that of Mol-Instructions [15], a dataset with well-curated human-annotated molecular tasks. These results indicate that data generated by ChemOrch effectively balances diversity and coverage, making it a competitive and diverse instruction dataset for chemistry learning.

**Generation Cost (token usage, expenses).** The instruction and response generation models are both based on LLMs. We analyze the generation cost in terms of token usage. Figure 4 shows the average token usage per generation, in different modules of ChemOrch. *Tool Selection* dominates the token usage (4094 tokens), followed by *Validation* (1472), *Answer Generation* (1165), and *Web Search* (1106). These four modules together consume the majority of tokens, indicating their central role in reasoning and verification. In contrast, modules like *Embedding Token* and *Self-Repairing* contribute minimally. This suggests that most token cost arises from tool planning and factual grounding rather

than lightweight utility steps. The total cost per instruction-response pair remains highly affordable, typically up to $0.05 per interaction, assuming the use of advanced reasoning models `o3-mini`[29]. Thus, even extensive reasoning processes involving multiple verification and generation steps are economically viable, highlighting the practical affordability of deploying ChemOrch.

Moreover, we also analyze the reasoning steps of generated responses in Appendix G.

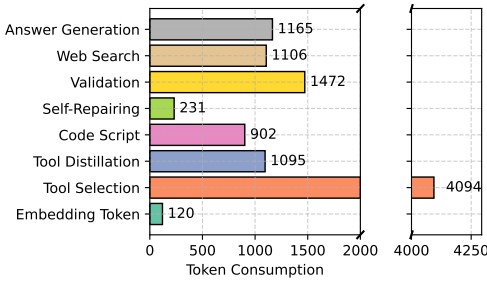

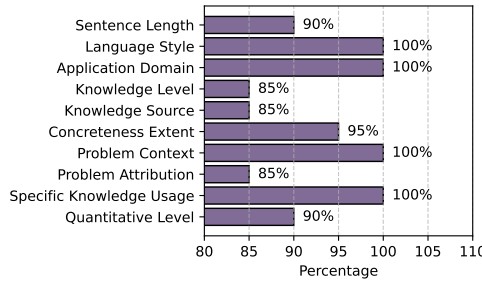

Figure 4: Token usage per generation in different modules of ChemOrch. "Answer generation" denotes the final assembly of the output.

Figure 5: Constraint-following accuracy evaluated by human annotator across 10 categories in instruction generation.

### 4.3  Quality Analysis of Generated Instruction–Response Pairs

**Human Evaluation of Response.** To assess the quality of the responses generated by ChemOrch, we conduct a human evaluation along two key dimensions: 1) *Instruction Following*–whether the response directly addresses the instruction; 2) *Factual Correctness*–whether the content is scientifically accurate. Details of the evaluation protocol and results of each task are provided in Appendix E. On average, the responses achieve an instruction following rate of 82.64% and a factual correctness rate of 85.14%. These results indicate that ChemOrch can generate responses that are aligned with user intent and maintain a high level of scientific reliability. Notably, the gap between instruction following and factual correctness suggests that factual errors are not the primary limitation; rather, improving instruction grounding, especially for complex multi-step queries, remains a potential area for enhancement. It is worth noting that integrating the tool pool significantly enhances ChemOrch's ability to generate factual responses. On name and property prediction tasks, ChemOrch achieves 75.00% accuracy, significantly outperforming GPT-4o's 22.50% without tool usage (see Appendix F). Moreover, we identify some failure modes of ChemOrch in human evaluation, providing some insights for future improvement, as detailed in Appendix K.

**Constraint Following.** To evaluate whether the instruction generation model adheres to user-specified constraints, we conduct a targeted constraint-following experiment. Specifically, we define 10 distinct categories of constraints, each encompassing multiple concrete constraint types. These categories cover aspects such as linguistic properties (e.g., sentence length, style), domain grounding (e.g., application domain, problem context), and knowledge control (e.g., knowledge level, concreteness, quantitative expression), as detailed in Appendix L.

To evaluate them, human annotators were asked to judge whether the generated instruction satisfies the specified requirement. The evaluation results are summarized in Figure 5. We observe that the instruction generation model demonstrates strong constraint alignment across most categories. In particular, it achieves 100% adherence in *language style*, *application domain*, *problem context*, and *specific knowledge usage*, indicating excellent controllability in terms of surface form and task semantics. The model also performs well on *sentence length* and *quantitative level* constraints, achieving 90% consistency. Notably, slightly lower scores are observed for *knowledge level*, *knowledge source*, and *problem attribution* (all at 85%), suggesting that while the model captures the intent of most constraints, nuanced knowledge-related instructions remain more challenging.

### 4.4  Application 1: Facilitating LLMs' Evaluation in Chemistry

In this section, we demonstrate the effectiveness of ChemOrch in facilitating both alignment of evaluation signals and the identification of weaknesses in LLMs on chemistry tasks. While many benchmarks have been proposed for evaluating the chemical abilities of LLMs [30, 11, 31, 32], constructing such datasets is often time-consuming and labor-intensive. ChemOrch offers a scalable

alternative by enabling the automatic generation of evaluation datasets for a given task, especially for those that are underrepresented, rarely addressed, or incorrectly handled in existing benchmarks. To assess ChemOrch's effectiveness in evaluation, we conduct two experiments: 1) Comparing LLM performance on ChemOrch-generated vs. existing benchmarks for consistency in trends; 2) Testing whether ChemOrch can expose LLM weaknesses in underrepresented chemical tasks. Additional experimental setup details are provided in Appendix B.

**1) ChemOrch vs. Existing Benchmarks: Aligning Evaluation Signals.** To assess whether ChemOrch is comparable to existing benchmarks, we select two chemical tasks, property prediction (PP) and molecular captioning (MC), with expert-curated evaluation samples from [31]. We run ChemOrch to generate 400 test samples for the same benchmarking tasks (used as few-shot examples as a constraint in ChemOrch), ensuring they are comparable in scope and difficulty to those in the benchmark. The results in Table 2 show that ChemOrch evaluation exhibits strong correlations with the original benchmarks: for PP, the Pearson correlation is $0.735$ with a $p$-value of $0.024$; for MC, the correlation is $0.948$ with a $p$-value less than $0.001$. While absolute error values vary across evaluated LLMs, their relative ranking remains largely preserved, indicating ChemOrch's consistency in comparative evaluation. This validates the reliability of ChemOrch, and highlights its potential for future use in scalable and adaptive evaluation.

**2) ChemOrch excels at revealing LLM weaknesses in chemistry by generating test data.** We identify several important chemical tasks that have not been widely covered in existing LLM benchmarks such as Lipophilicity Prediction. As shown in Table 3, we evaluate various models on samples generated by ChemOrch solely from the task metadata (from authoritative databases) and its description, without using any annotated examples. The results show that most models perform poorly on these tasks, revealing significant gaps in their chemical reasoning capabilities. This not only highlights the need for further improvement in LLMs but also demonstrates ChemOrch's effectiveness in identifying their weaknesses.

Table 2: The evaluation results (PP: Accuracy (0-1), MC: Score (1-5)) of two chemistry tasks on the original dataset (Ori.)[31] and generated dataset powered by ChemOrch (Ours).

| Model | Property Pred. | | Molecule Cap. | |
|---|---|---|---|---|
| | Ori. | Ours | Ori. | Ours |
| Llama-3.1-8B-Ins. | 0.203 | 0.051 | 3.28 | 3.52 |
| Qwen2.5-7B-Ins. | 0.580 | 0.277 | 4.54 | 5.34 |
| GPT-4o-mini | 0.418 | 0.292 | 5.52 | 6.56 |
| GPT-4o | 0.548 | 0.441 | 5.90 | 6.60 |
| Qwen3-14B | 0.593 | 0.549 | 5.66 | 6.60 |
| gemma-3-27b-it | 0.300 | 0.328 | 4.96 | 6.56 |
| DeepSeek-V3 | 0.450 | 0.523 | 5.92 | 6.96 |
| Llama-3.3-70B-Ins. | 0.470 | 0.364 | 4.96 | 6.14 |
| Claude3.5-haiku | 0.565 | 0.535 | 5.86 | 6.62 |

Table 3: Model performance (Accuracy) on fine-grained tasks in chemistry. B3D3 means Blood-Brain Barrier Penetration Prediction, DDI means Drug-Drug interaction.

| Model | B3D3 Prediction | DDI Prediction | Lipophilicity Prediction |
|---|---|---|---|
| Llama-3.1-8B-Instruct | $0.058_{\pm 0.010}$ | $0.000_{\pm 0.000}$ | $0.013_{\pm 0.000}$ |
| Qwen2.5-7B-Instruct | $0.236_{\pm 0.010}$ | $0.224_{\pm 0.010}$ | $0.036_{\pm 0.004}$ |
| GPT-4o-mini | $0.256_{\pm 0.010}$ | $0.176_{\pm 0.015}$ | $0.036_{\pm 0.004}$ |
| GPT-4o | $0.285_{\pm 0.020}$ | $0.133_{\pm 0.012}$ | $0.027_{\pm 0.007}$ |
| Qwen3-14B | $0.471_{\pm 0.004}$ | $0.347_{\pm 0.007}$ | $0.088_{\pm 0.010}$ |
| gemma-3-27b-it | $0.342_{\pm 0.004}$ | $0.520_{\pm 0.013}$ | $0.042_{\pm 0.007}$ |
| DeepSeek-V3 | $0.313_{\pm 0.018}$ | $0.280_{\pm 0.012}$ | $0.062_{\pm 0.004}$ |
| Llama-3.3-70B-Instruct | $0.133_{\pm 0.007}$ | $0.140_{\pm 0.000}$ | $0.036_{\pm 0.008}$ |
| Claude3.5-haiku | $0.287_{\pm 0.013}$ | $0.107_{\pm 0.007}$ | $0.056_{\pm 0.004}$ |

## 4.5 Application 2: Improving LLMs' Chemistry Intelligence

In this section, we demonstrate the effectiveness of the samples generated by ChemOrch in enhancing the chemistry capabilities of LLMs. ChemOrch is used to create diverse samples across a range of tasks, including task-specific questions, general Q&A, chemistry reasoning, and tool usage. We finetune Llama-3.1-8B-Instruct [33] and Qwen-2.5-7B-Instruct [34] by SFT on generated samples. More details are provided in Appendix B.

**Improvement on both the chemistry general-purpose Q&A and task-specific questions.** As shown in Figure 6, property prediction accuracy increased by around 35% for Llama-3.1-8B-Ins., indicating a stronger ability to infer molecular properties. For Molecule Captioning, scores improved from 0.5 to 1.2 for Llama-3.1-8B-Ins. and Qwen-2.5-7B-Ins., showing that models generated

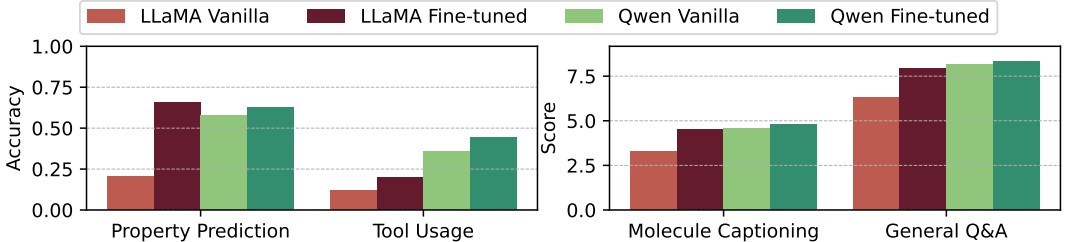

Figure 6: Fine-tuning results of Llama-3.1-8B-Instruct and Qwen2.5-7B-Instruct on tasks such as property prediction, molecule captioning, General QA and tool usage.

more accurate and informative descriptions. Similarly, in general, chemistry Q&A, we observed a comparable increase, suggesting enhanced domain understanding and reasoning ability. These results demonstrate that ChemOrch effectively boosts both specialized and general-purpose chemical Q&A.

**Improvement on the chemistry reasoning capability.** We evaluate how LLMs can improve their reasoning capability by fine-tuning on ChemOrch-generated data, promoted by chemistry reasoning questions from MMLU-Pro [32]. As shown in Table 4, both models show notable improvements over their vanilla baselines. For Llama-3.1-8B-Instruct, accuracy rises from 13.9% to 28.99% with 500 training samples. However, gains plateau beyond 200 samples, possibly due to the model's limited capacity or a distributional mismatch between ChemOrch data and Llama's generation style, which may hinder effective learning. In contrast, Qwen-2.5-7B-Instruct benefits more significantly from larger sample sizes, improving from 24.64% to 37.97%, suggesting stronger alignment with ChemOrch data and better instruction-following ability. Overall, these results demonstrate that ChemOrch can substantially enhance LLMs' chemistry reasoning capability.

**Improvement on the agentic capability.** The agentic framework has begun to receive attention within the chemistry domain [8], where tool-use capability is considered one of the most critical aspects of agentic models. To investigate whether ChemOrch can enhance a model's ability to use tools, we leveraged it to dynamically generate instructions and corresponding code snippets for two widely used chemistry tools: RDKit and PubChem. These generated examples were then used to fine-tune the model. As shown in Figure 6, the tool-use accuracy of the models improved significantly after fine-tuning. Notably, Llama-3.1-8B-Instruct achieved a relative improvement of over 50%, highlighting ChemOrch's effectiveness in facilitating tool-oriented skill acquisition.

| # Sample | Llama-3.1-8B-Ins. | Qwen-2.5-7B-Ins. |
|----------|-------------------|------------------|
| **Vanilla** | $0.1391_{\pm 0.0071}$ | $0.2464_{\pm 0.0041}$ |
| **n=200** | $0.2812_{\pm 0.0041}$ | $0.2870_{\pm 0.0123}$ |
| **n=300** | $0.2464_{\pm 0.0082}$ | $0.2928_{\pm 0.0147}$ |
| **n=400** | $0.2725_{\pm 0.0082}$ | $0.3478_{\pm 0.0213}$ |
| **n=500** | $0.2899_{\pm 0.0041}$ | $0.3797_{\pm 0.0041}$ |

Table 4: Model performance (Accuracy) on chemistry reasoning questions of MMLU-Pro [32] under different training sample sizes.

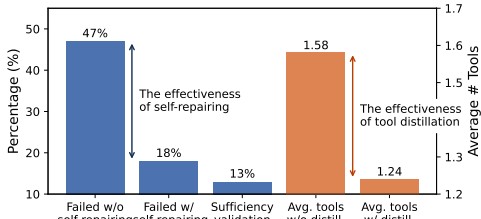

Figure 7: Ablation analysis of the core modules in ChemOrch.

## 4.6 Effectiveness of Core Modules in ChemOrch

The effectiveness of the difficulty reward model $\mathcal{M}_{\text{diff}}$ has been verified in Appendix D. The ablation studies of other modules including tool distillation, self-repairing, and sufficiency validation are shown in Figure 7. As we can observe, ablating each module in turn highlights its impact in a single sweep: disabling self-repairing causes the tool-execution failure rate to jump from 18% to 47%, adding sufficiency validation catches an extra 13% of incomplete or inadequate outputs, and applying tool distillation cuts the average number of tools invoked from 1.58 to 1.24 with no loss in overall success. These results indicate the effectiveness of core modules in ChemOrch.

### 4.7 Scalability of ChemOrch

ChemOrch supports scalable tools for handling more extensive chemistry tasks. For example, ChemOrch can support alternative molecular encodings, such as graph-based, tree-structured, or JSON representations by easily integrating a new tool function, which is detailed in Appendix O.

## 5 Conclusion

We introduce ChemOrch, a transformative framework for generating high-quality, tool-grounded instruction–response pairs in chemistry. It significantly lowers the barrier for assessing and further improving LLMs on chemistry tasks, particularly through the integration of chemical tools that enable accurate and verifiable reasoning. Its principles—task-conditioned generation, tool grounding, difficulty calibration, and repairability—are domain-agnostic and could help build cross-disciplinary AI models with expert-level abilities.

## Acknowledgement

This work was supported by the National Science Foundation under the NSF Center for Computer Assisted Synthesis (C-CAS), grant number CHE-2202693.

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

# Appendix Contents

# A Related Work

**Chemistry of LLMs.** LLMs are demonstrating remarkable capabilities within the chemistry domain. Key applications include predicting molecular properties[35, 14], generating novel molecular structures[36, 31, 37], and tackling complex problems in chemical synthesis and reaction informatics, such as planning reaction pathways and predicting outcomes[7, 38]. An emerging paradigm involves deploying LLMs as autonomous agents [39, 40], exemplified by systems like ChemCrow [41] and Coscientist [42], which integrate the LLM's reasoning capabilities with specialized external tools to automate complex scientific workflows. Despite this progress, significant challenges hinder LLMs from reaching their full potential in chemistry. Progress still stalls for three reasons: curated chemistry instructions are scarce, generic synthetic pipelines overlook molecular structure and domain rules, and existing generators struggle to produce diverse yet verifiable prompts and answers. These gaps motivate **ChemOrch**, which combines task-controlled instruction generation with tool-grounded, executable responses to yield large-scale, chemically sound training data.

**Synthetic data of LLMs.** LLMs have shown remarkable capabilities in generating synthetic data [43]. Unlike earlier approaches centered on traditional language models [44], latest LLMs offer greater potential for producing high-quality synthetic datasets across a variety of domains, including multilingual question answering [45], conversational agents [46], instruction tuning [18, 47, 48, 15], enhancing truthfulness [49], and promoting data diversity [50–53]. Most recently, DataGen [16] was introduced as a framework for generating high-quality textual datasets, enabling more targeted evaluation and improvement of LLM capabilities. Similarly, Lee et al. present Janus, an LLM trained on synthetically generated, diverse system messages to support personalized and general alignment [54]. Notably, Phi-4 [55] strategically incorporates synthetic data throughout the training process, which achieves an excellent performance on various downstream tasks.

# B Details of Experiment Setup

**Generation Details.** We include task descriptions in Table 5 and Table 6. For General Chemistry Q&A, we use the chemistry-related topics as constraints, as shown in Table 11 and Table 12. For task-specific challenges, we generate the dataset as follows:

- **Property Prediction & Molecule Captioning**: We leverage data samples from ChemLLMBench [11] as few-shot exemplars to guide the generation process of ChemOrch.
- **Tool Usage**: During generation, ChemOrch performs web searches to retrieve relevant code blocks or examples based on the instruction. To ensure correctness, the retrieved code is executed locally, and any code that results in errors is filtered out.
- **Chemistry Reasoning**: We use selected 115 examples from the chemistry reasoning questions in MMLU-Pro [32] (distinct from the evaluation set) as constraints (few-shot learning) to guide the model's generation. During this process, ChemOrch typically produces executable code for performing calculations and obtaining answers. To enable the synthesis of reasoning chains, we introduce an additional constraint that requires the model to generate code with printed intermediate results, thereby making the reasoning process explicit.
- **BBB Penetration Prediction & DDI Prediction & Lipophilicity Prediction**: We sample 200 seed data points for each task from three authoritative databases as the metadata for generation. The BBB penetration prediction comes from B3DB [56], the DDI prediction comes from TDC [57–59], and the lipophilicity prediction comes from MoleculeNet [60]. We transform the data into JSON format for each task as its metadata. Then, we send the metadata to both the IG and the RG models to generate accurate and reliable instruction-response pairs.

**Task Evaluation.** For evaluation, we adopt an LLM-as-a-Judge framework [61] across all tasks. Except for the molecule captioning task, where the LLM assigns a score from 1 to 5 by comparing the generated molecular description with the annotated description, as shown in Figure 34, all other tasks are evaluated by directly comparing the generated answers with the ground truth to determine correctness and reporting accuracy, as in Figure 33. For tasks involving tool usage, we similarly assess the correctness of the generated functional code block by comparing it to the ground truth implementation, treating it as a binary classification task, as in Figure 32.

**Data Used in Experiments.** For statistical analysis in subsection 4.2 and human evaluation in subsection 4.3, we randomly select 400 data points from both the General Chemistry Q&A and

Task-Specific Challenges datasets. To evaluate constraint adherence, we generate 100 instructions across 10 categories. The dataset used for the chemistry evaluation in subsection 4.4 contains 400 examples per task. For the fine-grained evaluation, each task includes 150 examples. For the fine-tuning experiments described in subsection 4.5, each of the three tasks (property prediction, tool usage, and molecule captioning) includes 400 samples for training and 400 for testing, with the test data sampled from the original benchmark [14]. For the general Q&A task, 1000 samples are used for training and 200 for testing–the larger size reflects the broader scope of chemical knowledge required. The ablation study of the ChemOrch module is conducted using a separate set of 200 data points. For all testing sets, we conduct a human evaluation to filter out low-quality data points.

**Hyperparameter Setting.** In our framework, we set a few hyperparameters to optimize the ability of our model. We set `top_k = 5` and `tool_distilling_num_threshold = 5` in the tool selection module to guarantee the selected tools' diversity and avoid tool redundancy. In the tool invacation module, we set `script_fixing_num_threshold = 3`, `error_fixing_num_threshold = 3`, and `effectiveness_checking_num_threshold = 5`. These settings ensure accurate code generation, which is closely related to the correctness of the results.

Table 5: Fine-Tuning Tasks

| Task Name | Description |
|---|---|
| Name Prediction | Predict the IUPAC name of the given molecular SMILES. |
| Property Prediction | Predict the property of molecules that the given reactants' SMILES represent. |
| Hydration free energy prediction | Predicts free energy of hydration of molecules, important for understanding solvation and interactions. |
| Molecule Captioning | Provide a detailed description of the molecule that the given molecular SMILES represents. |
| Reaction Prediction | Predict the main product SMILES according to the given reactants' SMILES. |

Table 6: Fine-grained Evaluation Tasks

| Task Name | Description |
|---|---|
| DDI Prediction | Predict the interaction type between two drugs. |
| BBB Penetration Prediction | Predict compounds' blood-brain barrier penetration. |
| Lipophility Prediction | Predict octanol/water distribution coefficient (logD) at pH 7.4. |

**Fine-Tuning Details.** We fine-tuned two instruction-tuned large language models: LLaMA-3.1–8B-Instruct and Qwen-2.5–7B-Instruct. All experiments were conducted using a consistent set of hyperparameters to ensure fair comparison between the models. Training was performed for 3 epochs with a cosine learning rate scheduler and a warmup ratio of 0.1. The learning rate was fixed at `1e-5`, and the per-device training batch size was set to 4, with no gradient accumulation (i.e., `gradient_accumulation_steps = 1`). We used bfloat16 (bf16) precision and trained on 4 NVIDIA A100 GPUs to accelerate computation and reduce memory usage. This setup provides an efficient and reproducible baseline for instruction tuning of large-scale language models.

**Constraint Generation.** Specifically, we prompt the LLM to generate concrete constraints under the broader constraint categories listed in Table 13. All the generated constraints are collected and then sequentially substituted into the instruction generation process of the IG model. In total, we collected over 100 different constraints to ensure diversity in the generated instructions.

# C   Details of Manual Review on Tool Decomposition

All sub-tools undergo a two-stage manual review. First, domain experts validate the input/output specifications, naming conventions, and descriptions to ensure alignment with chemical standards and usability. Second, reviewers simulate tool usage under realistic settings, including edge cases such as malformed molecular inputs or ambiguous return types. Through this process, we iteratively refine sub-tool definitions, add robust error handling, and rewrite unclear descriptions—ensuring that each sub-tool is both syntactically valid and semantically reliable.

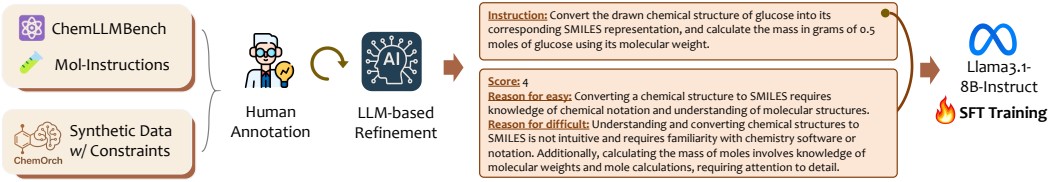

Figure 8: Data collection and training process of the *difficulty reward model with feedback*.

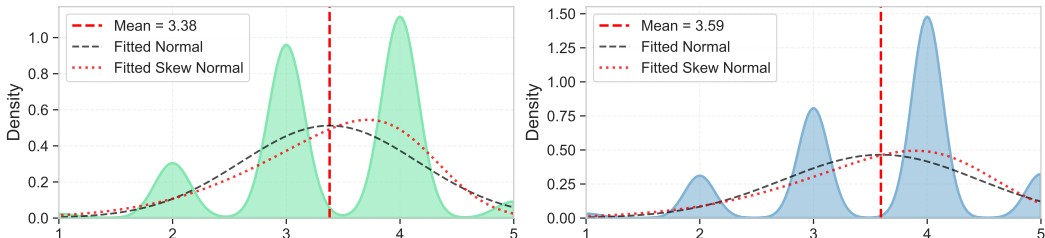

Figure 9: Data distribution comparison of test (left) and train (right) dataset.

The manual review process was conducted by a team of **nine experts** with diverse academic backgrounds to ensure both computational and chemical correctness. Specifically, the team comprised three undergraduate students and six PhD students. Among them, three undergraduates and two PhD students had a background in computer science, while the remaining four PhD students specialized in computational chemistry. This interdisciplinary composition ensured that each function was reviewed effectively.

# D  Details of the *difficulty reward model with feedback*

**Training data collection.** To train the difficulty reward model $\mathcal{M}_{\text{diff}}$, we construct a dataset comprising both synthetic and human-annotated instructions. The synthetic portion is generated using ChemOrch framework, covering a diverse range of chemistry tasks with variation introduced through a wide set of constraints, including both manually designed templates and LLM-generated constraint prompts. To complement this, we incorporate real instructions from existing datasets such as `Chem-LLMBench` [11] and `Mol-Instructions` [15], which are manually annotated by a team of three experts major in computational chemistry. Each instruction is labeled with a difficulty score on a 1–5 scale, along with textual explanations indicating the reasons for its simplicity and difficulty. To improve clarity and consistency, annotators use GPT-4o to refine their drafted annotations, helping to standardize language and eliminate ambiguity without altering the core assessment. Finally, we collect 3390 annotated data items.

**Training details.** We then train $\mathcal{M}_{\text{diff}}$ using supervised fine-tuning (SFT) to jointly predict the difficulty score and generate the corresponding feedback. The Meta-Llama-3.1-8B-Instruct is trained on 4×A100 for 3 epochs with a per-device batch size of 2, gradient accumulation of 1, and bf16 precision. The training employed the AdamW optimizer with a learning rate of $1 \times 10^{-5}$ and a cosine decay schedule. We used LLaMA-Factory [62] for the training process.

**Effectiveness evaluation.** The effectiveness evaluation is performed on a test instruction dataset of 900 samples, whose data distribution is tightly aligned with the training data of 3390 samples, as depicted in Figure 9, ensuring consistency across datasets. In our human evaluation protocol, each predicted difficulty score is presented to a computational chemistry expert alongside the corresponding instruction and the textual explanation. The expert then judges whether the score and reasoning are appropriate

Table 7: Human alignment rate of $\mathcal{M}_{\text{diff}}$.

| Difficulty Score | Human Alignment Rate |
|---|---|
| 1 | 100% |
| 2 | 88.9% |
| 3 | 85.7% |
| 4 | 86.8% |
| 5 | 100% |
| Total | 87.0% |

---

**Human Evaluation Guideline**

**Objective**
The objective of this validation task is to assess the quality of model-generated responses based on their corresponding instructions. Specifically, your role is to determine whether the response **appropriately addresses the instruction** and is **factually accurate**.

**Files and Format**
You will be provided with a JSON file containing a list of examples. Each example is represented as a dictionary with multiple fields. For this task, please focus exclusively on the following two:

- `instruction`: The user-provided prompt or question.
- `response`: The answer generated by the model.

Your task is to evaluate the quality of each response with respect to the instruction.

**Evaluation Criteria**
For each example, assign one of the following labels:

- **1 (Pass)**: The response accurately and sufficiently answers the instruction, with no factual errors.
- **0 (Fail)**: The response fails to address the instruction, or contains factual inaccuracies, hallucinations, or misleading content.
- **N/A (Uncertain)**: You are unable to confidently determine the quality of the response, due to ambiguity, insufficient domain knowledge, or unclear instruction.

---

Figure 10: Human evaluation guideline of response quality.

or not. Table 7 summarizes the effectiveness evaluation results, showing the alignment rate between $\mathcal{M}_{\text{diff}}$ and human expert judgments.

$\mathcal{M}_{\text{diff}}$ demonstrates a robust overall human alignment rate of 87%, indicating that the model successfully captures the nuances of chemistry instruction complexity. Difficulty levels 1 and 5 achieved perfect alignment (100%), while levels 2, 3, and 4 showed high alignment rates of 88.9%, 85.7%, and 86.8%, respectively. These results confirm that the model accurately reflects human judgment across varying complexity levels.

## E   Human Evaluation Details For Response Quality

**Evaluation Guideline.** The human evaluation guideline is shown in Figure 10. To ensure the reliability and professionalism of the assessment, the evaluation was conducted over a total of 400 samples by four Ph.D. students with backgrounds in computational chemistry. The guideline instructs annotators to determine whether the model-generated response (i) appropriately addresses the given instruction and (ii) contains no factual errors. Each response is labeled as **1 (Pass)**, **0 (Fail)**, or **N/A (Uncertain)**, depending on its accuracy and relevance. Annotators are instructed to focus solely on the `instruction` and `response` fields from each data entry. For the final analysis, samples labeled as **N/A** were excluded to ensure statistical validity.

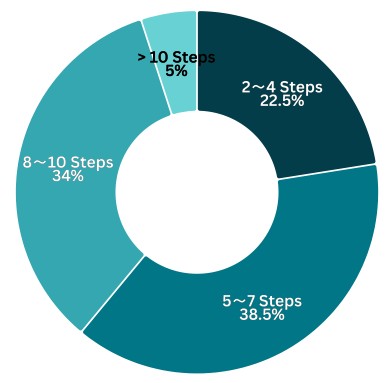

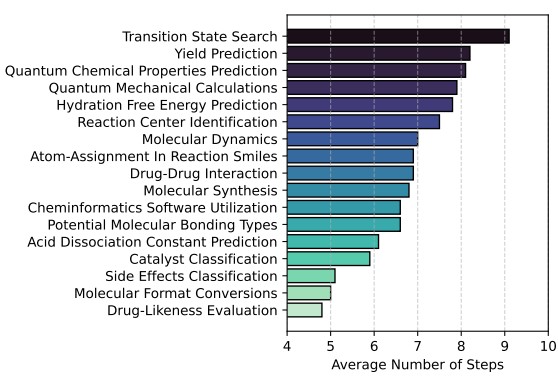

(a) The distribution of step number in generated responses.

(b) The step number in generated responses of different topics.

Figure 11: Statistics of reasoning step numbers in generated responses.

# F    Baseline Comparison

We collect 120 reliable metadata items from PubChem [20] and TDC [57–59] and send them to the IG model to generate the name prediction and property prediction task instructions. Then, we generate the responses using ChemOrch and the baseline model (GPT-4o [23]) for these tasks.

To evaluate the correctness of the generated pairs, we adopt LLM-as-a-Judge [61] to calculate the accuracy of ChemOrch and the baseline model. Our framework illustrates a high accuracy rate of 75.00%, which is significantly higher than the 22.50% of the baseline model.

# G    Reasoning Steps of Generated Responses

We automatically parse and get the number of steps in each response by GPT-4o. As illustrated in Figure 11, ChemOrch-generated responses exhibit a wide range of reasoning depths, with over 70% requiring more than five intermediate steps. This indicates that our framework promotes non-trivial, multi-stage reasoning beyond template-based generation. Moreover, the average number of steps varies substantially across topics, from more than nine steps for quantum-level predictions to fewer than six for format conversions, reflecting the framework's adaptive planning capability. These results validate the core design of ChemOrch: it is able to support diverse, complex, and execution-grounded reasoning chains at scale.

# H    Reliability of *LLM-as-a-Judge* Evaluation

To evaluate the quality and correctness of model-generated outputs across various tasks, we adopt the *LLM-as-a-Judge* paradigm, which leverages large language models to assess generated responses. Given the growing use of this evaluation strategy in recent literature, it is important to establish its empirical reliability, particularly in the absence of clear rule-based or human-labeled ground truth for complex tasks.

While rule-based metrics remain appropriate for simple binary classification, they may introduce inaccuracies in semantic evaluation (e.g., by failing to match semantically equivalent responses that differ lexically). The LLM-as-a-Judge approach offers broader applicability by capturing contextual nuances and aligning better with human preferences.

To assess the validity of this approach, we conducted a human–LLM agreement study across three tasks using two representative instruction-tuned models: `Llama-3.1-8B-Instruct` and `Qwen-2.5-7B-Instruct`. As shown in Table 8, Table 9, and Table 10, the results indicate strong alignment between automated scoring and human judgment: 1) **Binary classification (Property Prediction)**: agreement up to 99.75%; 2) **Binary classification (Tool Usage)**: agreement up to 97%; 3) **Score-based evaluation (General QA)**: average Pearson correlation $r = 0.796$ (all statistically significant). These results support the robustness of LLM-based evaluators as proxies for human judges in large-scale evaluation pipelines.

Table 8: Human–LLM agreement on property prediction task.

| Model | Batch 1 | Batch 2 | Avg. Alignment |
|---|---|---|---|
| Llama-3.1-8B-Instruct (1) | 49/50 (98%) | 50/50 (100%) | 99% |
| Llama-3.1-8B-Instruct (2) | 50/50 (100%) | 50/50 (100%) | 100% |
| Qwen-2.5-7B-Instruct (1) | 50/50 (100%) | 50/50 (100%) | 100% |
| Qwen-2.5-7B-Instruct (2) | 50/50 (100%) | 50/50 (100%) | 100% |
| **Overall Average** | — | — | **99.75%** |

Table 9: Human–LLM agreement on tool usage task.

| Model | Batch 1 | Batch 2 | Avg. Alignment |
|---|---|---|---|
| Llama-3.1-8B-Instruct (1) | 47/50 (94%) | 46/50 (96%) | 95% |
| Llama-3.1-8B-Instruct (2) | 47/50 (94%) | 50/50 (100%) | 97% |
| Qwen-2.5-7B-Instruct (1) | 48/50 (96%) | 50/50 (100%) | 98% |
| Qwen-2.5-7B-Instruct (2) | 48/50 (96%) | 50/50 (100%) | 98% |
| **Overall Average** | — | — | **97%** |

# I    Details of Selected Tools

ChemOrch leverages two categories of tools: chemistry-related tools such as RDKit [19] and PubChem [20] and general-purpose tools like web search.

**RDKit.** We utilize RDKit[2][19]–a widely adopted open-source cheminformatics toolkit–for molecular representation and processing during data synthesis. RDKit provides essential functionalities for SMILES parsing, molecular graph construction, substructure matching, and descriptor computation, which are critical for generating chemically valid and structurally diverse input-output instruction pairs. Its seamless integration with Python and support for 2D/3D molecular operations make it particularly suitable for large-scale instruction generation in the chemistry domain.

**PubChem.** We also incorporate data from PubChem[3] [20], a public repository maintained by the National Institutes of Health (NIH), which provides comprehensive information on chemical compounds, including their molecular structures, properties, bioactivities, and identifiers. PubChem serves as a reliable source for curating chemically diverse and biologically relevant compounds.

**Web Search.** We leverage the web search tool provided via the OpenAI API[4] to retrieve up-to-date and domain-relevant information from the internet in real time. This enables our framework to enrich instruction data with factual context and emerging knowledge beyond the model's pretraining corpus.

**Reasons for Selection of PubChem and RDKit.** We select PubChem and RDKit as the foundational tools in ChemOrch due to their broad functionality, stable APIs, and suitability for function-level decomposition. While alternative toolkits such as DeepChem [63] and ASKCOS [64] are widely used in the chemistry community, they present practical limitations in the context of instruction-based tool invocation. DeepChem [63], for instance, emphasizes model training and evaluation pipelines, requiring users to manage datasets, train predictors, and interpret model outputs. This training-heavy workflow is often too heavy-weight for lightweight, step-level function calling and lacks the immediacy and transparency needed for modular LLM usage. ASKCOS [64] provides powerful capabilities in retrosynthesis and reaction planning but is optimized for end-to-end synthesis tasks and requires complex orchestration or server-side APIs, making it difficult to extract self-contained functions for flexible invocation.

In contrast, RDKit offers atomic-level cheminformatics operations (e.g., SMILES parsing, substructure matching, fingerprinting) with lightweight and stable interfaces. PubChem provides robust access to curated compound data and chemical identifiers through scalable and open APIs. They strike a

---

[2]https://www.rdkit.org/
[3]https://pubchem.ncbi.nlm.nih.gov/
[4]https://platform.openai.com/docs/guides/tools-web-search

Table 10: Human–LLM agreement on score-based evaluation (General QA).

| Model | Pearson $r$ |
|---|---|
| Llama-3.1-8B-Instruct (1) | 0.741 |
| Llama-3.1-8B-Instruct (2) | 0.728 |
| Qwen-2.5-7B-Instruct (1) | 0.859 |
| Qwen-2.5-7B-Instruct (2) | 0.854 |
| **Average** | **0.796** |

balance between expressiveness, modularity, and integration ease–making them ideal building blocks for constructing function-level primitives in ChemOrch.

Table 11: Molecular property prediction topics.

| Topic Name | Description |
|---|---|
| Partition coefficient prediction | This task involves predicting the partition coefficient (log P) of molecules, which reflects their hydrophobicity and is crucial for understanding their pharmacokinetic properties. |
| Water solubility prediction | This task involves predicting the solubility of compounds in water, using datasets like ESOL. |
| Hydration free energy prediction | Predicts free energy of hydration of molecules, important for understanding solvation and interactions. |
| Lipophilicity prediction | Predicts tendency to dissolve in lipids, a measure of lipophilicity. |
| Quantum chemical properties prediction | Predicts quantum chemical properties, such as energies and geometries, using QM7, QM8, and QM9 datasets. |
| Blood–Brain Barrier Penetration prediction | The task involves predicting molecules' Blood-Brain Barrier penetration capability. |
| Protein–Ligand Binding Affinity prediction | Predicts binding affinity in terms of Kd (dissociation constant) using PDBbind data. |
| BACE Inhibition prediction | Predicts compound's ability to inhibit BACE protein. |
| HIV Inhibition prediction | Predicts compound's ability to inhibit HIV replication. |
| Side Effect prediction | Predicts the side effects of drugs across multiple categories. |
| Drug–Drug Interaction prediction | Whether two drugs will interact, and their interaction type. |
| Clearance prediction | Forecasts the clearance rate of compounds from biological systems. |
| Oral Bioavailability prediction | Determines whether a molecule is orally available or not (or has prodrugs). |
| Enzyme Interaction prediction | Identifies which enzyme(s) a drug inhibits. |
| pKa prediction | The task involves the estimation of the acid dissociation constant (pKa) of molecules. |

# J  Broader impacts

ChemOrch represents a transformative advancement at the intersection of LLMs and computational chemistry. By generating high-quality, tool-grounded instruction–response data at scale, ChemOrch lowers the barrier for training and evaluating LLMs on chemistry tasks. This capability has particular value for researchers and institutions with limited access to curated chemical datasets, helping to democratize access to domain-specific tools powered by generative AI.

**ChemOrch can accelerate scientific discovery.** It equips LLMs with structured chemical reasoning abilities, which can support innovation in areas such as drug discovery, materials design, and reaction informatics [65]. Researchers can use ChemOrch-generated tasks to identify model weaknesses, construct fine-tuning datasets, and conduct more rigorous benchmarking. These capabilities can lead to faster hypothesis testing, reduced experimental costs, and more informed scientific decisions.

**ChemOrch can support education and workforce development.** The system can generate chemistry problems and reasoning tasks with adjustable difficulty, making it suitable for instructional use. Students can interact with AI-generated content to deepen their understanding of complex topics, while educators can tailor assignments to various learning levels. This flexibility makes ChemOrch especially useful in educational settings with limited access to expert instructors or resources.

Table 12: Reaction-level prediction topics.

| Topic Name | Description |
| --- | --- |
| Retrosynthetic Analysis | The task involves determining feasible starting materials and stepwise synthesis routes for a target molecule. |
| Reaction Type Classification | This task involves classifying the type of chemical reaction. |
| Reaction Center Identification | This task involves pinpointing the specific site(s) in a molecule where reaction occurs. |
| Reaction Condition Recommendation | This task involves recommending optimal reaction conditions (solvent, catalyst, etc.). |
| Solvent Classification | The task involves categorizing solvents by their chemical properties. |
| Ligand Classification | This task involves categorizing and distinguishing different ligands. |
| Catalyst Classification | This task involves sorting catalysts into different classes. |
| Reaction Temperature Prediction | This task involves predicting the optimal temperature for a given reaction. |
| Reactant Amount Prediction | This task involves predicting the required quantities of reactants. |
| Reaction Time Prediction | This task involves estimating the reaction duration. |
| Reaction Workup Recommendation | This task involves proposing procedures for post-reaction purification. |
| Yield Prediction | The task involves estimating the amount of product formed. |
| Selectivity Prediction | This task involves predicting the selectivity between possible products. |
| Reaction Outcome Prediction | This task involves predicting the outcomes of a reaction given reactants and conditions. |
| Reaction Outcome Rationalization | This task involves explaining why a particular reaction outcome occurs. |
| Stereoselectivity Prediction | This task involves analyzing the preferential formation of stereoisomers. |

**ChemOrch can enable chemistry-aware autonomous agents.** By integrating task decomposition, tool execution, error correction, and difficulty calibration, ChemOrch provides the foundation for building LLM-based agents capable of operating in scientific domains [66, 8, 67]. These agents could assist in experiment planning, molecular analysis, and literature synthesis—enhancing collaboration between human researchers and AI models.

# K   Limitations and Failure Modes for the Future Work

Despite the strong performance of ChemOrch in synthesizing domain-specific instructions for chemical reasoning, our in-depth analysis reveals several unique and non-trivial failure modes that highlight opportunities for further improvement:

**Conflict Between Tool Outputs and Model Knowledge.** We observe that factual inaccuracies in some responses stem not only from tool malfunctions but more subtly from a mismatch between tool outputs and the model's internal knowledge. In certain cases, the model ignores the tool's returned result–especially when it contradicts its prior knowledge or learned biases [68]. This suggests that the model does not always treat the tool as a trusted authority. A promising direction to mitigate this issue is to incorporate an explicit instructional priority hierarchy [69], in which tool outputs are assigned a higher trust level than model-generated content, encouraging the model to defer to tools in cases of conflict.

**Error Cascades Due to Incorrect Tool Usage.** We identify a failure pattern in which an early-stage tool invocation error–such as supplying an invalid or malformed SMILES string–propagates through subsequent steps, resulting in entirely flawed reasoning chains. These snowballing errors highlight the brittleness of current tool integration. A potential solution involves developing more robust error detection and rollback mechanisms, allowing the system to identify and correct invalid tool inputs before proceeding with subsequent reasoning steps.

**Model Laziness in Complex Instructions.** For particularly complex instructions, we find that the model often resorts to generating high-level guidance (e.g., "You can use PubChem to search...") rather than executing the task and providing a concrete answer. This "lazy" behavior may be an artifact of underlying system prompts used in alignment-tuned models (e.g., OpenAI's usage constraints), which prioritize efficiency and safety over exhaustive computation. Future work could explore prompt-level interventions or model fine-tuning strategies to better incentivize execution over delegation.

**Planning and Step Ordering Errors.** A notable failure mode arises from logical inconsistencies in the generated reasoning plans–such as incorrect ordering of steps or violations of necessary

chemical dependencies (e.g., attempting a reaction analysis before retrieving the molecular structure). These issues reflect a fundamental challenge in planning for domain-specific tasks, where procedural correctness is tightly coupled with chemical constraints. While much of the current progress in LLM-based reasoning has focused on general domains, our observations underscore the importance of domain-adapted reasoning capabilities tailored for chemistry. Future directions may include the integration of hierarchical planning modules and chemistry-aware workflow decomposition [8], which explicitly model task-specific execution order and causal dependencies.

## L Constraint Examples

We show the constriant examples of different aspects in Table 13.

Table 13: Constraint examples of instruction generation.

| Constraint category | Example |
|---|---|
| Sentence Length | Use extremely concise sentences, limited to 5-10 words, retaining only the most essential information. |
| Language Style | Employ a humorous and lighthearted tone with anthropomorphic or whimsical analogies. |
| Application Domain | Explore physical chemistry problems related to thermodynamics/kinetics calculations. |
| Knowledge Level | Tailor content for elementary students using only common-sense descriptions. |
| Knowledge Source | Reference recent findings from top-tier journal publications within three years. |
| Concreteness Extent | Maintain completely abstract descriptions without concrete examples. |
| Problem Context | Contextualize within industrial production line scenarios. |
| Problem Attribution | Formulate mechanism analysis questions with electron-pushing arrows. |
| Specific Knowledge Usage | Involve titration equivalence calculations or endpoint determination. |
| Quantitative Level | Develop mathematical models or algorithmic optimization requirements. |

## M Algorithm of ChemOrch

---

**Algorithm 1** Instruction Generation

---

**Require:** Task space $\mathcal{T}$, Constraint set $\mathcal{C}$, Metadata set $M$
**Require:** Instruction Generation model $\mathcal{M}_{\text{inst}}$, Difficulty reward model $\mathcal{M}_{\text{diff}}$
 1: **for** each task $t \in \mathcal{T}$ **do**                          ▷ Iterate over all tasks
 2:     **for** each constraint $c \in \mathcal{C}$ **do**                  ▷ Iterate over all constraints
 3:         **for** each metadata $m \in M$ **do**              ▷ Iterate over all metadata
 4:             $x \leftarrow \mathcal{M}_{\text{inst}}(t, c, m)$              ▷ Generate initial instruction
 5:             $(d, e) \leftarrow \mathcal{M}_{\text{diff}}(x)$           ▷ Evaluate difficulty and feedback
 6:             **while** difficulty $d$ does not meet target level **do**   ▷ Repeat if difficulty is misaligned
 7:                 $x \leftarrow \mathcal{M}_{\text{inst}}(t, c, m, e)$           ▷ Regenerate using feedback
 8:                 $(d, e) \leftarrow \mathcal{M}_{\text{diff}}(x)$             ▷ Re-evaluate difficulty
 9:             **end while**
10:             Save or store final instruction $x$                  ▷ Store final instruction
11:         **end for**
12:     **end for**
13: **end for**

---

**Algorithm 2** Response Construction with Tool-Driven Execution

---

**Require:** Instruction $x$, Metadata $m$, Tool pool $\mathcal{F}$, Response generation model $\mathcal{M}_{\text{resp}}$
1: $s \leftarrow \text{Decompose}(x)$           $\triangleright$ Decompose instruction into reasoning steps
2: $\{d_1, \ldots, d_M\} \leftarrow \mathcal{M}_{\text{resp}}(s, m)$           $\triangleright$ Predict expected tool descriptions
3: **for** each $d_m \in \{d_1, \ldots, d_M\}$ **do**
4:      $\mathcal{F}_m^{\text{raw}} \leftarrow$ Top-$k$ retrieved tools via cosine similarity    $\triangleright$ Semantic retrieval of candidate tools
5: **end for**
6: $\mathcal{F}^{\text{raw}} \leftarrow \bigcup_{m=1}^M \mathcal{F}_m^{\text{raw}}$           $\triangleright$ Aggregate raw candidate tool pool
7: $\mathcal{F}^\star \leftarrow \text{Distill}(\mathcal{F}^{\text{raw}}, x, s, m, \tau)$        $\triangleright$ Refine tools via usefulness, expertise, and budget
8: **for** each $f \in \mathcal{F}^\star$ **do**
9:      $\mathcal{S}_f \leftarrow \mathcal{M}_{\text{resp}}(f, x, s, m)$           $\triangleright$ Generate code script for tool $f$
10:      Execute $\mathcal{S}_f$; if success, store output $o_f$           $\triangleright$ Run script and store result
11:      **if** execution fails **then**
12:          **for** attempt $i = 1$ to $R_{\max}$ **do**
13:              Capture error $e$, regenerate $\mathcal{S}_f^{(i+1)} \leftarrow \mathcal{M}_{\text{resp}}(e, \mathcal{S}_f^{(i)}, m)$     $\triangleright$ Self-repair using error trace
14:              Retry execution
15:              **if** success **then**
16:                  Store output $o_f$; **break**
17:              **end if**
18:          **end for**
19:          **if** still failed **then**
20:              Retrieve external documentation and regenerate $\mathcal{S}_f$   $\triangleright$ Fallback to external guidance
21:          **end if**
22:      **end if**
23:      Check result sufficiency for $o_f$; refine if needed           $\triangleright$ Validate quality of tool output
24:      Check if $\{o_f\}$ satisfies instruction $x$; stop early if sufficient   $\triangleright$ Early stopping for efficiency
25: **end for**
26: **if** final outputs insufficient **then**
27:      $o_{\text{extra}} \leftarrow \text{WebSearch}(x, m)$           $\triangleright$ Trigger web search for missing information
28: **end if**
29: $\mathcal{O} \leftarrow \{o_f\}_{f \in \mathcal{F}^\star} \cup \{o_{\text{extra}}\}$           $\triangleright$ Aggregate all outputs
30: $y \leftarrow \mathcal{M}_{\text{resp}}(x, \mathcal{O}, m)$           $\triangleright$ Construct final response grounded in outputs

---

# N    Data Examples and Case Study

To provide a clearer understanding of model behavior in ChemOrch, we present examples of the model's output for *Instruction Decomposition* (Figure 31) and *Expected Tool Specification* (Figure 14), using the prompt: "How can Lipinski's Rule of Five be used to assess the drug-likeness of a compound?". Moreover, we provide examples of generated instruction-response pairs for tasks including molecule caption, property prediction, general chemistry Q&A, chemistry reasoning, and tool usage in Figure 15, Figure 16, Figure 17, Figure 18, and Figure 19.

Supplementing our primary experiments, we conducted additional fine-tuning using data generated by ChemOrch on two downstream tasks, solvent classification and catalyst classification. As shown in Figure Figure 12, both tasks achieved performance gains. These results further highlight the efficacy of ChemOrch.

To demonstrate the extensibility of ChemOrch, we present a representative use case involving a customized tool, as illustrated in Figure 20.

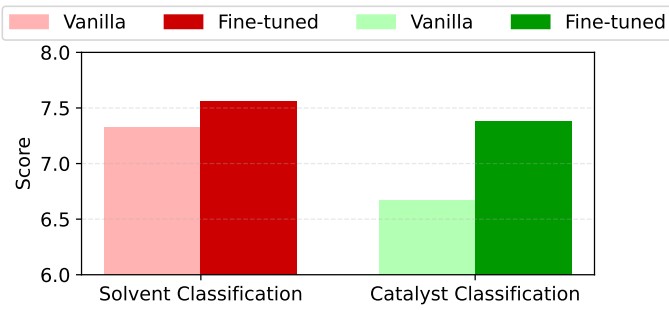

Figure 12: Fine-tuning results on two tasks: solvent classification and catalyst classification

**Instruction Decomposition**

Step1: Research and summarize Lipinski's Rule of Five, focusing on its criteria for drug-likeness.
Step2: Identify the key parameters of Lipinski's Rule of Five: molecular weight, logP, hydrogen bond donors, and hydrogen bond acceptors.
Step3: Acquire the chemical structure of the compound to be assessed for drug-likeness.
Step4: Calculate the molecular weight of the compound using its chemical structure.
Step5: Determine the compound's partition coefficient (logP) to evaluate its hydrophobicity or hydrophilicity.
Step6: Count the number of hydrogen bond donors (e.g., NH or OH groups) in the compound's structure.
Step7: Count the number of hydrogen bond acceptors (e.g., N or O atoms) in the compound's structure.
Step8: Compare the calculated values against Lipinski's criteria: molecular weight < 500 Da, logP < 5, hydrogen bond donors < 5, and hydrogen bond acceptors < 10.
Step9: Assess the compound's drug-likeness based on its conformity to Lipinski's Rule of Five.
Step10: Consider using cheminformatics software tools (e.g., ChemDraw, RDKit) for automated calculations and analysis.

Figure 13: Example of instruction decomposition by GPT-4o.

**Expected Tool Specification**

Molecular weight calculator: Computes the molecular weight of a compound from its chemical structure.
LogP calculator: Determines the partition coefficient of a compound to assess its hydrophobicity or hydrophilicity.
Hydrogen bond donor counter: Counts NH and OH groups in a compound's chemical structure.
Hydrogen bond acceptor counter: Counts nitrogen and oxygen atoms in a compound's chemical structure.
Cheminformatics structure viewer: Visualizes the chemical structure of a compound.

Figure 14: Example of expected tool specification by GPT-4o.

## Molecule Caption Task

**Instruction:** Can you provide a descriptive analysis of the molecule that corresponds to the SMILES: C1=CC(=CC=C1NC(=O)C2=CC=C(O2)C3=C(C=CC(=C3)Cl)Cl)I?

**Response:** The molecule corresponding to the SMILES string C1=CC(=CC=C1NC(=O)C2=CC=C(O2)C3=C(C=CC(=C3)Cl)Cl)I has the molecular formula C17H10Cl2INO2 and a molecular weight of 458.082 g/mol. It contains several functional groups, including an amide group signified by the NC(=O) structure, an ether group within an aromatic system, and halogens represented by chlorine (Cl) and iodine (I) atoms. The molecule is highly aromatic, featuring three interconnected aromatic rings. The first aromatic ring is a phenyl unit C1=CC(=CC=C1), while the second C2=CC=C(O2) contains an ether linkage, making it an aryl ether. The third aromatic ring C3=C(C=CC(=C3)Cl)Cl contains two chlorine atoms. Overall, this compound is a halogenated aromatic molecule with complex characteristics due to its amide, ether functionalities, and halogen substituents, which provide potential sites for chemical reactivity or interaction. Its structure indicates potential diverse chemical behavior due to the multiple aromatic systems and functional groups.

Figure 15: Example of molecule caption task from ChemOrch.

## Property prediction Task

**Instruction:** Assess if the compound with SMILES: C1=CC=CC=C1C2=CC=CC=C2 can penetrate the blood-brain barrier.

**Response:** Yes. Biphenyl, represented by the SMILES notation C1=CC=CC=C1C2=CC=CC=C2, demonstrates physicochemical properties that suggest it can penetrate the blood-brain barrier (BBB). Its molecular weight of 154.21 g/mol, $\log P$ of approximately 4.0, polar surface area (PSA) of $0\text{Å}^2$, and lack of hydrogen bond donors or acceptors all fall within the favorable ranges for BBB penetration. Biphenyl's non-polar nature and absence of functional groups also suggest minimal interaction with active transport mechanisms, further indicating its potential for BBB penetration.

Figure 16: Example of property prediction task from ChemOrch.

## General Chemistry Question Task

**Instruction:** What is the role of catalysis in the Haber process for ammonia production?

**Response:** The role of catalysis in the Haber process for ammonia production is to significantly enhance the efficiency of the reaction by increasing the reaction rate, facilitating the reaction mechanism, optimizing operational conditions, and improving industrial significance. Catalysts lower the activation energy required for nitrogen and hydrogen to react, with iron being a typical catalyst. This enables the reaction to proceed rapidly at feasible temperatures and pressures. The catalyst also verifies adsorption of reactants, weakening the nitrogen triple bond to facilitate the formation of ammonia, while maintaining energy efficiency and cost-effectiveness, crucial for large-scale production.

Figure 17: Example of general chemistry question task from ChemOrch.

**Instruction:** At 15°C, a solution exhibits a pH value of 2.73. Calculate the corresponding pOH and the hydroxide-ion concentration, $[\text{OH}^-]$.

**Response:** To solve this problem, we want to determine both the pOH of the solution and the concentration of hydroxide ions, $[\text{OH}^-]$, at a temperature of 15°C, given that the solution has a pH of 2.73. Here's how you can approach the solution step by step:

1. **Understand the Relationship Between pH, pOH, and pKw:**
   The pH of a solution is a measure of its acidity, while pOH is a measure of its basicity. The two are related through the ion product of water, $K_w$, especially since we know the temperature-dependent value of $K_w$ at 15°C.

2. **Using pH and pKw to Find pOH:**
   At any given temperature, there's a relationship between pH, pOH, and pKw described by the equation:
   $$\text{pH} + \text{pOH} = \text{pKw}$$
   At 15°C, we know that $\text{pKw} = 13.94$. You're also provided with a specific pH of 2.73 for the solution. So, to find pOH, we rearrange the equation to:
   $$\text{pOH} = \text{pKw} - \text{pH} = 13.94 - 2.73$$
   By performing this calculation, we determine:
   $$\text{pOH} = 11.21$$

3. **Understanding the Relationship Between pOH and $[\text{OH}^-]$:**
   The pOH value gives us insight into the concentration of hydroxide ions in the solution. The relationship between pOH and the concentration of $\text{OH}^-$ ions is described by:
   $$\text{pOH} = -\log_{10}\big([\text{OH}^-]\big)$$
   In order to find $[\text{OH}^-]$, we need to rearrange this equation to solve for the concentration:
   $$[\text{OH}^-] = 10^{-\text{pOH}}$$

4. **Calculating the $[\text{OH}^-]$ Concentration:**
   Now, with the pOH we calculated as 11.21, we can determine the concentration of hydroxide ions:
   $$[\text{OH}^-] = 10^{-11.21} \approx 6.12 \times 10^{-12}\,\text{M}$$

This final value tells us the concentration of hydroxide ions present in the solution at $15\,^{\circ}\text{C}$, rounding off our solution process using the given inputs. By following these logical steps, we not only confirm the veracity of the result but also deepen our understanding of the relationships between pH, pOH, and the ion product of water at specific temperatures.

Figure 18: Example of chemistry reasoning task from ChemOrch.

**Instruction:** Utilize the *get_compounds* function from the PubChemPy library to find the chemical details for a compound with the SMILES string CC[N+](C)(CC)CCC(C)(C#C)O and set smiles as the identifier namespace.

**Response:**

```python
try:
    from pubchempy import get_compounds

    # define the parameters
    identifier_value = "cc[n+](c)(cc)ccc(c)(c#c)o"
    namespace_value = "smiles"

    # call the get_compounds function with the provided
        parameters
    result = get_compounds(identifier=identifier_value, namespace
        =namespace_value)

    # print the result with an explanatory message
    print("the chemical details for the compound with the smiles
        string '{}' are: {}".format(identifier_value, result))
except Exception as e:
    print("error: {}".format(e))
```

Figure 19: Example of tool usage task from ChemOrch.

**Customized tool example**

**User customized tool information:** *smiles_from_compound*

```
additional_tool = [{
    "tool": "smiles_from_compound",
    "module": "ord_schema.message_helpers",
    "description": "Fetches or generates a SMILES identifier for
        a compound. If a SMILES identifier already exists, it is
        simply returned.",
    "parameters": {"compound": "reaction_pb2.Compound message."},
    "documentation": "https://docs.open-reaction-database.org/en/
        latest/ord_schema/ord_schema.html#module-ord_schema.
        message_helpers"
}]
```

**Tool calling script:**

```python
import ord_schema.message_helpers
from ord_schema import reaction_pb2
try:
    # Create a Compound object and add a SMILES identifier for
        ethanol
    compound = reaction_pb2.Compound()
    identifier = compound.identifiers.add()
    identifier.type = reaction_pb2.CompoundIdentifier.SMILES
    identifier.value = "CCO"

    # Call the function from the module with the properly formed
        compound object
    result = ord_schema.message_helpers.smiles_from_compound(
        compound)
    print("The SMILES string for ethanol is:", result)
except Exception as e:
    print("Error:", e)
```

Figure 20: Example of customized extended tool.

## O   Scalability Example

Importantly, ChemOrch can be readily extended to handle alternative molecular formats by simply introducing appropriate conversion functions in the preprocessing stage. For example, to support graph-based representations, one only needs to add a transformation module before the main task. Below, we provide a code snippet illustrating how ChemOrch can seamlessly convert a graph representation:

```python
def graph_to_iupac_name(graph):
    mol = Chem.RWMol()
    atom_idx_map = {}

    # Add atoms
    for i, atom_info in enumerate(graph["atoms"]):
        atom = Chem.Atom(atom_info["element"])
        atom.SetFormalCharge(atom_info.get("charge", 0))
        atom.SetIsAromatic(atom_info.get("is_aromatic", False))
        idx = mol.AddAtom(atom)
        atom_idx_map[i] = idx

    # Add bonds
    bond_order_map = {
        "single": Chem.rdchem.BondType.SINGLE,
        "double": Chem.rdchem.BondType.DOUBLE,
        "triple": Chem.rdchem.BondType.TRIPLE,
        "aromatic": Chem.rdchem.BondType.AROMATIC
    }

    added = set()
    for a1, neighbors in graph["bonds"].items():
        for a2, bond_type in neighbors:
            if (a2, a1) in added:
                continue
            bt = bond_order_map.get(bond_type.lower())
            if bt is None:
                raise ValueError(f"Unknown bond type: {bond_type}")
            mol.AddBond(atom_idx_map[a1], atom_idx_map[a2], bt)
            added.add((a1, a2))

    mol.UpdatePropertyCache(strict=False)
    Chem.SanitizeMol(mol)
    return pcp.get_compounds(Chem.MolToSmiles(mol, canonical=True),
        'smiles')[0].iupac_name
```

# P    Prompt Template

---
**Instruction Synthesis Prompt**

You are an advanced AI assistant tasked with generating high-quality instructions for synthetic dataset creation.

Your goal is to produce a diverse set of instructions (or questions) based on a given user task. The corresponding answers will be generated later to form a dataset.
### **Instructions:**
1. **Task Understanding:** Carefully analyze the provided task and determine its core objective.
2. **Instruction Generation:** Create exactly 'n' unique instructions related to the task. The instructions should be diverse in phrasing and complexity.
3. **Clarity & Context:** Ensure each instruction is clear and provides enough context for an AI model to generate a meaningful response.
4. **Format:** Return the instructions strictly as a Python-style list of strings.
5. **Custom Constraint:** {custom_constraint}
6. **Metadata:** If metadata is provided, your instructions should adhere to it.
### **Example:**
#### **User Task:** Toxicity Prediction
#### **Generated Instructions (Example Output):**
[
"Does benzo[a]pyrene exhibit toxicity to humans?",
"What is the acute toxicity of trichloroethylene?",
"Does bisphenol A have endocrine-disrupting effects?",
"Do pyridine compounds have neurotoxic effects?",
"Does tetraethyl lead pose long-term toxicity risks to the environment and humans?"
]

---

Figure 21: Instruction synthesis prompt for ChemOrch.


### **Instructions:**
1. **Understand the Instruction:** Carefully analyze the given instruction to determine its requirements.
2. **Identify Key Elements:** Identify key components such as subject, method, and expected output.
3. **Break Down into Steps:** Generate a structured plan consisting of logical steps that guide the problem-solving process.
4. **Ensure Tool Compatibility:** If an external tool is likely required (e.g., a chemical database, scientific literature, mathematical solver), indicate it explicitly.
5. **Format:** Return the planning steps strictly as a Python-style list of strings.
6. **Metadata:** If metadata is provided, your planning should centre on it.
Now, generate a structured plan for the following instruction:
Instruction: {instruction}
Ensure the output is formatted strictly as a Python list of strings.

Figure 22: Instruction decomposition prompt for ChemOrch.

You are an advanced AI assistant tasked with defining the ideal tools for executing a plan.
Your goal is to describe the functionalities of these tools concisely, ensuring that each tool serves **one specific purpose**.
### **Instructions:**
1. **Analyze the Planning Steps:** Carefully review the provided planning steps to determine what kind of external tools would be needed to complete them.
2. **Define the Ideal Toolset:** Describe **only the necessary** tools, ensuring that each tool performs only **one function**.
3. **Keep Descriptions Concise:** Each tool description should be brief and focused on its function.
4. **Limit the Number of Tools:** Minimize the number of tools by **combining related functionalities** into single tools where applicable.
5. **Format:** Return the tool descriptions strictly as a Python-style list of strings.
6. **Metadata:** If metadata is provided, your tool planning should refer to it.
Now, generate a structured list of ideal tool descriptions for the following planning steps:
Planning Steps: {planning_steps}
Ensure the output is formatted strictly as a Python list of strings, with each tool description containing only one function.

Figure 23: Tool planning prompt for ChemOrch.

acknowledge that the true answer is often more nuanced, so please just use your best judgment and write a justification to elaborate. All supporting evidence can appear either in the main paper or the supplemental material, provided in appendix. If you answer [Yes] to a question, in the justification please point to the section(s) where related material for the question can be found.

IMPORTANT, please:

- **Delete this instruction block, but keep the section heading "NeurIPS Paper Checklist",**

- **Keep the checklist subsection headings, questions/answers and guidelines below.**

- **Do not modify the questions and only use the provided macros for your answers**.

**Tool Retrieval Prompt**

I will give you the task, a tool name, and its description.
Your goal is to confirm whether the tool can be used to solve the task.
Instructions:
1. You need to extract the final targets of the task and determine whether it requires a specific tool or multiple tools.
2. First, you need to focus on solving the final targets of the task.
3. Second, if the task requires multiple tools and this tool excels in one aspect of the task, it is also useful.
4. If metadata is provided, your choice of tool should be based on the requirements of the metadata.
Output format:
1. If the tool can be used for solving the task, return the tool index only. It should be an integer.
2. If the tool can't be used for solving the task, return the string "no" only. It should be a string.

Figure 24: Tool retrieval prompt for ChemOrch.

**Tool Distillation Prompt**

I will give you a list of tools that have been screened, and they are all related to the task. I will also give you the raw task.
Problems:
1. Although these tools are all related to the task, some may be indirectly related to the task, or the tool may not be an expert in the task.
2. Some tools may not be able to solve the final targets of the task.
Your goal is to check the tools and confirm whether they need to remove some indirectly related tools.
Strategies for tool selection:
1. Pay attention to the tools' names. The tool name contains its function, and if the task needs the tool, the name often appears in the tool description.
2. Throw light on the task content. The content may clarify what tools or what kinds of tools are needed for the task.
Instructions:
1. Read the tools list and the task carefully, compare the tools' functions with the task, and check if the task marks specific tools to use.
2. Analyse the task and extract the final targets of the task. Regarding the tools can't solve the final targets of the task as useless tools, you should focus on the final targets of the task.
3. If the number of tools overnumbers the threshold:{threshold_for_tool_distilling}, you should think more about finding and removing indirectly related tools. In another situation, if the task only needs a few steps to solve, you should think more about using fewer tools.
4. If metadata is provided, your choice of tool should be based on the requirements of the metadata.
Output format:
1. If the indirectly related tools are found, please return only the most indirectly related tool index.
2. If no indirectly related tools are found, please return the string "no" only.
3. You should return the content described above without any prefixes or suffixes.

Figure 25: Tool distillation prompt for ChemOrch.

**Code Script Generation Prompt**

I will give you some key-value pairs that describe the task, module name, function name, and parameters with specific values.
Your goal is to write a script for calling the function with the given parameters.
Instructions:
1. Import the module in this format:
"import ChemGen.tools.module_name" or "import module_name".
The module name will be given in the user prompt under the "module_name" key.
2. Some parameters may need other packages. Please check the parameters and import the required packages.
3. Create variables for the parameters and fill them with the given values.
4. Call the function with the parameters and print the result. When printing the result, you need to describe what it means and not just print it.
Important:
The function name will be in the user prompt under the "function name" key.
Output format:
Return the script content only without any useless prefixes or suffixes.

Figure 26: Code script generation prompt for ChemOrch.

**Self-Repairing: Error Catching Prompt**

I will give you a Python script and its error message.
Your goal is to fix the error in the script according to the error message.
Output format:
Return the fixed script content only, without any useless prefixes or suffixes like double quotation or back quote marks to mark this as a Python file.

Figure 27: Error catching prompt for ChemOrch.

**Self-Repairing: Effectiveness Checking Prompt**

I will give you the task, the planning steps for solving the task, the script for the task, and its output.
Your goal is to determine whether the output is useful for solving the task.
The criteria for judging the uselessness of the output:
1. The output is an object without valid characters or numeric information. This one is important and often appears. Please pay attention.
2. The output is discordant or irrelevant to the task.
3. The script does not follow the planning steps, focusing on checking the input variables and output format.
4. The output is not the accurate data the task requires.
If you find the output is useless, you can modify the script according to the website given below:
{website}
Output format:
1. Return the "useful" string only if the script output is useful.
2. Return the modified script content only if the output script is useless.
3. The modified script content should be without any useless prefixes or suffixes like double quotation or back quote marks.

Figure 28: Effectiveness checking prompt for ChemOrch.

**Sufficiency Validation Prompt**

I will give you a task and the results of some tools used to solve the task.
Your goal is to judge whether the present results are sufficient for solving the task.
Output format:
1. Return the string "yes" only if the results are sufficient.
2. Return the string "no" only if the results are insufficient.

Figure 29: Sufficiency validation prompt for ChemOrch.

**Web Search Prompt**

I will give you a task and the planning steps for solving the task.
Your goal is to search for the related information to solve the task online.

Figure 30: Web search prompt for ChemOrch.

**Answer Assembly Prompt**

I will give you a task and some information generated from some tools for the task.
Your goal is to analyze and solve the task. You can choose useful information generated from the tools to make your answer accurate and correct.
Instructions:
1. Read the task carefully and analyze its requirements.
2. Read the information given by the tools carefully and determine whether it can be used directly.
3. If the information cannot be used directly, you should transform it according to the task's requirements.
4. If you receive multiple answers but they are different, you can process them in two ways:
(1) Choose the most accurate answer based on your judgment.
(2) If the answers have descriptions about how they are generated, you can output all answers with their descriptions and let the user choose the most accurate one.
5. Ensure the answer has good readability. You can change the illustration format if needed.

Figure 31: Answer assembly prompt for ChemOrch.

**LLM-as-a-Judge Evaluation Prompt**

You are an evaluator for grading the quality of answers to chemistry questions. I will provide you with a question, a predicted answer, and a reference answer.
Your task is to compare the predicted answer to the reference answer and assess how well they match in meaning.
Consider factors such as accuracy, completeness, and clarity, even if the wording is different. Please first analyze the differences and similarities between the predicted and reference answers.
Then give a final score from 1 to 10, where 10 means a perfect match in meaning and 1 means completely incorrect.
Here is the question: [[Question]]
Here is the predicted answer: [[Answer]]
Here is the reference answer: [[Label]]
Respond only with the following format on the final line:
Final score: X

Figure 32: LLM-as-a-Judge evaluation prompt for molecule captioning

---
**LLM-as-a-Judge Evaluation Prompt**

You are an evaluator tasked with assessing whether a model's code-based solution to a problem is correct.
You will be given a problem statement, the model's generated code (predicted answer), and a reference solution (correct answer).
Your job is to determine whether the model's code is functionally or logically equivalent to the reference solution. Please carefully compare the predicted code with the reference code, analyze their logic and behavior, and finally respond with either 'correct' or 'incorrect'.
Problem: [[Question]]
Predicted Code: [[Answer]]
Reference Code: [[Label]]

---

Figure 33: LLM-as-a-Judge evaluation prompt for tool usage

---
**LLM-as-a-Judge Evaluation Prompt**

You are an evaluator for evaluating whether a response to a chemistry question is correct or not.
I will provide you with a question, the predicted answer, and the correct answer.
Your task is to determine if the predicted answer matches the correct answer in meaning, even if the wording is slightly different. Please first compare the predicted answer with correct answer and analyze them, and finally respond with 'correct' or 'incorrect'.
Here is the question: [[Question]]
Here is the predicted answer: [[Answer]]
Here is the correct answer: [[Label]]

---

Figure 34: LLM-as-a-Judge evaluation prompt for other tasks

