# OpenReview forum: "ChemOrch: Empowering LLMs with Chemical Intelligence via Groundbreaking Synthetic Instructions"
_NeurIPS.cc/2025/Conference — NeurIPS 2025 poster_

### Official Review · Reviewer_4t1k · 2025-06-24

**Clarity:** 3
**Significance:** 3
**Originality:** 2
**Rating:** 3
**Confidence:** 4

**Summary:**

The paper introduces ChemOrch, a two-stage framework that generates chemical instruction–response pairs to address the shortage of high-quality domain data for LLMs. Comprehensive experiments show the results have higher diversity. LLMs trained on the ChemOrch show improved capabilities.

**Questions:**

Q1. What types of tools are specifically required for Chemistry Q&A tasks? Given that existing work has shown LLMs can already perform reasonably well on such questions, what additional capabilities do these tools provide, and in what ways do they further enhance the performance of LLMs?

**Ethical Concerns:**

["NO or VERY MINOR ethics concerns only"]

**Final Justification:**

I appreciate the author’s efforts in addressing the comments. However, my concern remains insufficiently addressed. I am still unclear on how generating synthetic data without any accuracy guarantees can be considered acceptable. In fact, the results presented by the authors appear to reinforce this concern. Therefore, I intend to maintain my current score.

**Limitations:**

Yes

**Quality:**

3

**Strengths And Weaknesses:**

S1. The paper is clearly motivated by three concrete gaps, including data scarcity, mismatch between generic synthetic pipelines and chemical rules, and lack of controllable diversity.

S2. The framework yields inexpensive, verifiable data.

S3. Multiple metrics are used to show the effectiveness of the proposed method.

W1. My primary concern lies in the accuracy of the generated instruction–response pairs. The current framework does not appear to guarantee correctness in generation. If the synthesized data contains factual errors or inconsistencies, it raises doubts about using it to fine-tune the LLM model.

W2. The early stopping mechanism is not clear to me. It is unclear why terminating execution early is preferable when the final outputs may still be incorrect.

W3. Certain parts of the evaluation rely heavily on human judgment.  This may introduce potential bias.

---

> ### Author Rebuttal · Authors · 2025-07-28
>
> We sincerely thank you for the thoughtful and detailed feedback. Below we respond to each point individually.
>
> ------
>
> **W1. Concern about the factual accuracy of generated instruction–response pairs.**
>
> **A1:** Thank you for raising this important concern. We respectfully believe there may have been a misunderstanding about the nature of our data generation pipeline.
>
> To clarify: **ChemOrch is explicitly designed to \*avoid\* relying on free-form LLM outputs for response generation**, precisely **because of the risk of hallucinations and factual errors**. Instead, we adopt a **tool-grounded generation framework**, where responses are produced through structured **tool calls** executed by trusted chemistry engines such as **RDKit** and **PubChem**.
>
> This approach ensures that outputs are not hallucinated, but rather **scientifically derived from verified computations and databases**. We further apply multiple safeguards—including **error detection**, **self-repairing logic**, and **sufficiency validation** (Sections 3.2–3.3)—to verify the completeness and correctness of generated responses before they are accepted.
>
> In addition, we conduct **extensive human evaluation** to empirically validate the quality of instruction–response pairs. As reported in Section 4.3 and Appendix G:
>
> - ChemOrch-generated data achieves **82.64% instruction-following accuracy**
> - And **85.14% factual correctness**, as judged by domain experts
>
> These results, along with a transparent annotation protocol and clearly reported sample sizes, offer **strong empirical support** that the generated data is of high quality and appropriate for downstream fine-tuning.
>
> We hope this clarifies the design intent of ChemOrch and reassures you that factual accuracy is both a **central motivation** and a **validated strength** of our framework.
>
> ------
>
> **W2. Unclear motivation behind the early stopping mechanism.**
>
> **A2:** Thank you for highlighting this point. We apologize for the lack of clarity.
>
> The **early stopping mechanism** in ChemOrch is not used to skip or prematurely terminate response generation. Instead, it is purely a **computational efficiency optimization**, applied **only after the system determines that all instruction requirements have been successfully satisfied**.
>
> To ensure no part of the instruction is overlooked, early stopping is always followed by a **sufficiency validation** step. If this check identifies any missing or incomplete outputs, execution resumes and additional tool calls are triggered.
>
> In short:
>
> - Early stopping is **never applied at the expense of correctness**
> - It only eliminates **redundant computation** when a task is verifiably complete
>
> We will clarify this mechanism further in the revised manuscript.
>
> ------
>
> **W3. Concerns about human judgment introducing bias in evaluation.**
>
> **A3:** We appreciate your thoughtful observation. We address this concern through a robust, multi-pronged evaluation framework combining both human annotation and LLM-as-a-Judge validation.
>
> ---
>
>  **1. Human Evaluation Protocol – Designed to Minimize Bias**
>
> We adopt several measures to reduce subjectivity:
>
> - **Fact-based assessment**: Annotators were encouraged to consult authoritative chemical resources (e.g., PubChem, textbooks) when uncertain.
> - **Diverse reviewer pool**: Our evaluation involved **nine annotators** (3 undergraduates + 6 PhD students) from ML, chemistry, and chemical engineering backgrounds.
> - **Blind and randomized assignment**: Evaluation was conducted blindly across models and batches to avoid anchoring bias.
>
> To further validate the internal consistency of our human evaluations, we computed the **inter-annotator agreement** using **Cohen’s Kappa coefficient** across three separate batches. As shown below, the Kappa scores indicate **substantial to almost perfect agreement**:
>
> | Batch   | Kappa Score |
> | ------- | ----------- |
> | Batch 1 | 0.862       |
> | Batch 2 | 0.623       |
> | Batch 3 | 0.722       |
>
> These results demonstrate that human raters maintained a high degree of consistency across tasks, further supporting the reliability of our human evaluation protocol.
>
> ---
>
> **2. Necessity of Human Evaluation in Open-Domain Tasks**
>
> We respectfully argue that human evaluation is indispensable for open-domain chemistry tasks, for the following reasons:
>
> 1. **Automatic metrics are insufficient**: Metrics like BLEU, ROUGE, or embedding similarity only assess surface-level overlap. They cannot reliably detect serious scientific errors, such as incorrect property values, invalid tool usage, or misapplied reaction mechanisms.
>
> 2. **Chemistry tasks require expert reasoning**: Many tasks involve multi-step logic, adherence to domain-specific rules, and scientific justification. These qualities are hard to quantify using purely automated metrics, especially for open-ended or complex instructions.
>
> 3. **Human evaluation remains the gold standard**: As supported by prior work [1–4], expert human judgment remains essential for evaluating synthetic data and LLM performance in scientific domains. In our study, annotators relied on structured criteria and external references to ensure objectivity.
>
> ---
>
> **3. LLM-as-a-Judge (High Alignment with Human Judgment)**
>
> To further validate reliability, we conducted a human–LLM agreement study across three tasks and two strong models (`LLaMA-3.1-8B-Instruct` and `Qwen-2.5-7B-Instruct`).
>
> - **Binary alignment (Property Prediction)**: up to **99.75%**
> - **Binary alignment (Tool Usage)**: up to **97%**
> - **Score-based correlation (General QA)**: average **Pearson r = 0.7955**, all **statistically significant**
>
> These results confirm strong agreement between automated and human judgments. Summary tables below:
>
> ---
>
> **Binary Accuracy – Property Prediction**
>
> | Task                | Model                     | Batch 1      | Batch 2      | Avg. Alignment |
> | ------------------- | ------------------------- | ------------ | ------------ | -------------- |
> | Property Prediction | LLaMA-3.1-8B-Instruct (1) | 49/50 (98%)  | 50/50 (100%) | 99%            |
> |                     | LLaMA-3.1-8B-Instruct (2) | 50/50 (100%) | 50/50 (100%) | 100%           |
> |                     | Qwen-2.5-7B-Instruct (1)  | 50/50 (100%) | 50/50 (100%) | 100%           |
> |                     | Qwen-2.5-7B-Instruct (2)  | 50/50 (100%) | 50/50 (100%) | 100%           |
> | **Overall Average** | —                         | —            | —            | **99.75%**     |
>
> ---
>
> **Binary Accuracy – Tool Usage**
>
> | Task                | Model                     | Batch 1     | Batch 2      | Avg. Alignment |
> | ------------------- | ------------------------- | ----------- | ------------ | -------------- |
> | Tool Usage          | LLaMA-3.1-8B-Instruct (1) | 47/50 (94%) | 46/50 (96%)  | 95%            |
> |                     | LLaMA-3.1-8B-Instruct (2) | 47/50 (94%) | 50/50 (100%) | 97%            |
> |                     | Qwen-2.5-7B-Instruct (1)  | 48/50 (96%) | 50/50 (100%) | 98%            |
> |                     | Qwen-2.5-7B-Instruct (2)  | 48/50 (96%) | 50/50 (100%) | 98%            |
> | **Overall Average** | —                         | —           | —            | **97%**        |
>
> ---
>
> Together, our inter-annotator agreement, expert-informed annotation protocol, and strong human–LLM alignment provide strong evidence for the robustness and fairness of our evaluation process.
>
> ---
>
> #### **2. Score-based Evaluation: General Q&A**
>
> | Model                     | Pearson r | P-value      |
> | ------------------------- | --------- | ------------ |
> | Llama-3.1-8B-Instruct (1) | 0.741     | 4.82 × 10⁻¹⁶ |
> | Llama-3.1-8B-Instruct (2) | 0.728     | 1.88 × 10⁻¹⁶ |
> | Qwen-2.5-7B-Instruct (1)  | 0.859     | 9.31 × 10⁻²⁸ |
> | Qwen-2.5-7B-Instruct (2)  | 0.854     | 2.20 × 10⁻²⁸ |
> | **Average**               | **0.796** | —            |
>
> These **high correlations (r ≈ 0.73–0.86)** with highly significant p-values demonstrate that our LLM judge is consistent with human judgments even in open-ended tasks.
>
> ------
>
> **Q1. What specific tools are required for Chemistry Q&A tasks, and how do they enhance LLM performance?**
>
> **A4:** Thank you for this insightful question.
>
> While LLMs can handle many chemistry-related questions reasonably well, they often fall short when dealing with:
>
> - **Uncommon or specialized topics**
> - **Out-of-date information**
> - **Facts that require external validation**
>
> To address this, ChemOrch integrates **web retrieval** as the primary tool for Chemistry Q&A tasks. This allows the model to access up-to-date scientific knowledge from reliable sources, verify its outputs with external evidence to reduce hallucinations, and accurately handle rare or niche chemistry questions.
>
> We will expand the explanation of tool roles in Chemistry Q&A in the revised manuscript to make these enhancements more explicit.
>
> ------
>
> [1] Huang, Yue, et al. *"Datagen: Unified synthetic dataset generation via large language models."* ICLR 2024.
>
> [2] Bran, Andres M., et al. *"Augmenting large language models with chemistry tools."* Nature Machine Intelligence, 2024.
>
> [3] Long, Lin, et al. *"On LLMs-Driven Synthetic Data Generation, Curation, and Evaluation: A Survey."* ACL (Findings), 2024.
>
> [4] Mirza, Adrian, et al. "A framework for evaluating the chemical knowledge and reasoning abilities of large language models against the expertise of chemists." *Nature Chemistry* (2025): 1-8.
>
> ------
>
> Once again,  we deeply appreciate your careful review and constructive comments. If you feel that our responses have sufficiently addressed your concerns, we would be truly grateful if you would consider revisiting your rating. Thanks you!

---

> > ### Comment · Reviewer_4t1k · 2025-08-05
> >
> > I appreciate the authors' rebuttal.
> >
> > I am still concerned about the accuracy of the generated instruction–response pairs. I don't think there is any misunderstanding. Using tools can be better than free-form LLM outputs, but it doesn't guarantee correctness. The 82.64% instruction-following accuracy and 85.14% factual correctness also confirm my concern.

---

> ### Author Response · Authors · 2025-08-05
> **Thanks for your response.**
>
> We sincerely thank the reviewer for the thoughtful feedback. We believe there may be a potential misunderstanding regarding factual correctness in synthetic data generation: **even the most advanced frameworks are hard to achieve 100% accuracy**, regardless of domain.
>
> **R1. On the nearly impossibility of 100% correctness – even in general domains**
>
> State-of-the-art general-domain systems such as **DataGen** [1] also report non-negligible factual and executional errors. This limitation arises from intrinsic challenges such as ambiguous prompts and edge-case failures.
>
> In contrast, **chemical instruction generation is substantially more difficult**, as it involves precise validation of molecular structures, reaction pathways, and symbolic representations—often with no clear-cut gold standard. **Expecting 100% accuracy in this setting sets a bar not demanded in any comparable domain.**
>
> **R2. ChemOrch’s correctness is both high and conservatively measured**
>
> Despite this intrinsic difficulty, **ChemOrch achieves 85.14% factual correctness and 82.64% instruction-following accuracy**—measured under **strict expert-defined criteria**:
> - No partial credit for incomplete or partially correct answers
> - All property values, structures, and tool calls must exactly match trusted sources
> - Annotators referenced authoritative databases (e.g., PubChem) during scoring
>
> These thresholds are **stricter than prior work** in both chemistry and general-domain synthetic generation, making our results especially robust. As a comparison point, **baseline LLM generations in our setting achieve ≤63% factual correctness**.
>
> **R3. Variation across tasks is expected and inherent to difficulty**
>
> In our benchmark, certain tasks such as **SMILES conversion** achieve **near-perfect accuracy** because they can be solved deterministically via tool calls. The lower accuracy for the most challenging tasks is expected — even expert chemists may not consistently produce correct answers for these, due to intrinsic complexity, open-ended reasoning, and multiple valid solution paths.
>
> **R4. Diversity vs. correctness trade-off**
>
> Some prior work achieves higher reported accuracy by **restricting to a narrow set of easier tasks**  (e.g., reasoning tasks [2]), which inevitably sacrifices diversity and coverage. In contrast, **ChemOrch supports a wide spectrum of chemistry tasks**, including:
> - Deterministic structure/property lookups
> - Tool-augmented open-ended reasoning
> - Multi-step synthesis planning
> - Complex property prediction
>
> This broad coverage is essential for training LLMs to handle realistic, heterogeneous chemistry scenarios, even if it naturally lowers the global average accuracy.
>
> **R5. Correctness enforcement is built into ChemOrch’s design**
>
> In ChemOrch, we further incorporate:
> - **Error detection and self-repairing logic**
> - **Sufficiency validation before early stopping**
>
> This leads to **substantial error reduction** and makes ChemOrch one of the first pipelines to support **scalable, verifiable instruction synthesis for chemistry**.
>
> **R6. Practical effectiveness: correctness sufficient for LLM tuning**
>
> Most importantly, the correctness level achieved by ChemOrch has proven to be **sufficient and effective for downstream LLM fine-tuning**. As shown in Section 4.4, models trained on ChemOrch:
> - Outperform baselines across multiple tasks in current benchmarks
> - Exhibit improved tool usage and factual grounding
>
> ---
>
> [1] Huang, Yue, et al. "Datagen: Unified synthetic dataset generation via large language models." The Thirteenth International Conference on Learning Representations. 2024.
>
> [2] Zhu, Kaijie, et al. "DyVal: Dynamic Evaluation of Large Language Models for Reasoning Tasks." The Twelfth International Conference on Learning Representations.
>
>
> ---
>
> We sincerely respect your comment and fully acknowledge the importance of factual correctness in synthetic chemistry data generation. While 100% accuracy is inherently nearly unattainable in both general-domain and chemistry-specific pipelines, we have demonstrated that ChemOrch achieves state-of-the-art performance under strict evaluation standards, offers broad task coverage beyond what prior work supports, and delivers tangible downstream benefits. We would be truly grateful if, in light of these clarifications and evidence, you could kindly reconsider your evaluation of our work. Thank you!

---

### Official Review · Reviewer_XBcL · 2025-06-29

**Clarity:** 3
**Significance:** 3
**Originality:** 3
**Rating:** 4
**Confidence:** 4

**Summary:**

The submission "ChemOrch: Empowering LLMs with Chemical Intelligence via Groundbreaking Synthetic Instructions" describes a framework for automatic construction of instruction-response pairs for large language models tailored to chemistry tasks. This is important as the number of available datasets in this domain is very limited. The framework provides data of comparable quality to existing approaches relying on human supervision but of significantly higher diversity. The utility of the framework of the framework was demonstrated to evaluate existing large language models and finetune them for further performance improvement. Overall, the study is well written and provides a significant advance in the field. I think it can benefit from some further improvements that should be addressed before publication. Please find my detailed comments below.

**Questions:**

Major aspects:
When evaluating the capability of ChemOrch to be used for finetuning, it seems that ChemOrch is both used for generating training data and for performance evaluation. This could result in data leakage. I think it is important that at least one example is provided where ChemOrch is used for finetuning but performance on other datasets such as the ones that already exist in literature (ChemLLMBench, Mol-Instructions) is shown.

Recently, another benchmark named ChemBench was published (https://doi.org/10.1038/s41557-025-01815-x). I think at least some comparison of ChemOrch to ChemBench should be provided.

The model evaluation metrics should be explained explicitly in the main text as the text is otherwise very hard to follow. It should be mentioned at least what the best and worst performance metric values are for each metric.

Page 8: The authors state: "Similarly, in general, chemistry Q&A, we observed a comparable increase, suggesting enhanced domain understanding and reasoning ability." I do not agree that a better performance directly suggests enhanced domain understanding and reasoning ability. I think it merely suggests a higher chance to provide results that are closer to the reference values.

I think that the comparison to existing approaches is relatively minimal. One existing study that could possibly provide more tasks to compare ChemOrch to previously tested tasks is ChemCrow (https://doi.org/10.1038/s42256-024-00832-8).

I think that more context should be provided regarding to human evaluation of response (section 4.3). I think it should be provided in the main text how many samples were evaluated in total and how the metrics was calculated. For instance, is an example with a single error as inaccurate as an example with many errors?

Based on Figure 6, it seems that the LLaMA model seems to benefit significantly more from finetuning than Qwen. Do the authors have an idea of the origin of this and the possible implications?

The submission mentions "Transision State Search" as one of the reasoning tasks. I could not find an example for this task in the Appendix. I think it is important to provide at least one example for each of the task categories to make it more clear to the reader what type of input-output pairs are considered for each of these.

Minor aspects:
Table 2: The evaluation results for MC are supposed to range between 1 and 5 (based on the table caption) but many results are even above 6. I think this is confusing and requires further clarification.

Figure 3: It would be insightful to compare the distrubtions to the reference datasets ChemLLMBench and Mol-Instructions.

**Ethical Concerns:**

["NO or VERY MINOR ethics concerns only"]

**Final Justification:**

I appreciate the efforts of the reviewers to address all my comments. I think most of them have been resolved satisfactorily. Therefore, I decided to raise my review score to 4. I think that ChemOrch is a clear advance in the field that will be helpful for large language model development in chemistry. For a higher score, I think a more comprehensive set of tasks relevant in the chemistry domain for model evaluation would be needed, especially with respect to more complex tasks that involve several step for their solution.

**Limitations:**

Yes.

**Paper Formatting Concerns:**

None.

**Quality:**

3

**Strengths And Weaknesses:**

I think the quality is fair as the framework fills an important niche in the field but it should carefully address concerns regarding data leakage.
I think the clarity is good as all the important components of the developed framework are illustrated well.
I think the significance is fair as the framework is demonstrated to be directly applicable to finetuning existing models and improve their performance significantly. However, it is important to evaluate its utility for improving the general capabilities of a model on other datasets as well.
I think originality is good as the framework is general and can be applied to many domains of science.

---

> ### Author Rebuttal · Authors · 2025-07-28
>
> We sincerely thank you for the detailed and thoughtful feedback. We address each point in detail below:
>
> **Q1: Concern about potential data leakage. Can you show evaluation on external datasets like ChemLLMBench or Mol-Instructions?**
>
> Thank you for raising this important concern. We apologize for any confusion and **respectfully clarify that this issue is already addressed in the manuscript (Lines 676–678)**:
>
> > “For the fine-tuning experiments described in subsection 4.5, each of the three tasks (property prediction, tool usage, and molecule captioning) includes 400 samples for training and 400 for testing, with the test data sampled from the original benchmark.”
>
> Specifically:
> - For **Property Prediction**, **Molecule Captioning** and **Tool Usage**, test sets are **fully sourced** from **ChemLLMBench**.
> - In the **Tool Usage** task, test prompts are derived from real-world function documentation (not model-generated), and molecules are randomly **sampled from ChemLLMBench**. Evaluation is based on exact output match, ensuring objectivity.
>
> Regarding the General Q&A, we did consider using subsets of MMLU and SciBench. However, the chemistry questions in these datasets are mostly focused on mathematical calculations, with relatively few items testing core chemical knowledge. While we acknowledge that our current Q&A set is generated by ChemOrch (as a necessary compromise), we have nonetheless applied data filtering to ensure quality, as stated in Lines 680–681:
>
> > “For all testing sets, we conduct a human evaluation to filter out low-quality data points.”
>
> This filtering follows the rigorous protocol in Appendix E. We will clarify this in the revision to ensure full transparency of our evaluation pipeline.
>
> ---
>
> **Q2: Comparison to ChemBench**
>
> Thank you for this excellent suggestion. While ChemBench is a valuable benchmark for systematically evaluating LLMs’ chemical knowledge and reasoning, our ChemOrch framework offers several clear advantages:
>
> 1. **Scalable data generation**: ChemOrch automatically produces diverse, high-quality chemistry instruction–response pairs at scale, without manual labeling.
> 2. **Broader and practical task coverage**: It includes not only Q&A but also complex tasks like property prediction and tool-based reasoning with executable outputs.
> 3. **Targeted for model improvement**: ChemOrch can generate data tailored to specific weaknesses, supporting both evaluation and fine-tuning.
>
>
> We now compared **ChemOrch** to **ChemBench**, **ChemLLMBench**, and **Mol-Instructions** on two diversity metrics:
>
> | Dataset             | APS ↓ | Remote-Clique ↑ |
> | ------------------- | ----- | --------------- |
> | **ChemOrch (ours)** | 0.779 | 0.661           |
> | ChemLLMBench        | 0.884 | 0.453           |
> | Mol-Instructions    | 0.765 | 0.683           |
> | ChemBench           | 0.784 | 0.613           |
>
> These metrics indicate that ChemOrch achieves **higher instruction diversity than ChemBench**.
>
> ---
>
> **Q3: The evaluation metrics are unclear. Please define them clearly in the main text.**
>
> Thank you for pointing this out. We agree that metric definitions are crucial for clarity. Below are the metrics we use, now explicitly described in the revised manuscript:
>
> - **Property Prediction Accuracy**: Proportion of correctly predicted properties (range: 0–1). Higher is better.
> - **Tool Usage Accuracy**: Based on whether tool outputs match expected function calls (range: 0–1). Higher is better.
> - **Molecule Captioning**: Score on a **1–10 scale**. Higher is better.
> - **General Chemistry Q&A**: Score on a **1–10 scale**. Higher is better.
> - **Instruction Following Rate / Factual Correctness Rate**: Percentage of responses rated by humans as fully instruction-compliant or scientifically accurate (range: 0–1).
> - **Diversity Metrics**:
>   - **Average Pairwise Similarity (APS)**: Lower is better.
>   - **Remote-Clique Score**: Higher indicates broader semantic coverage.
>
>
> ---
>
> **Q4: Better performance on chemistry Q&A doesn't necessarily prove domain understanding or reasoning.**
>
> Thank you for the thoughtful critique. We agree that **higher benchmark scores do not automatically imply true domain understanding or reasoning**.
>
> Our original phrasing—“suggesting enhanced domain understanding”—may overstate this link. In the revision, we now clarify that:
>
> > “The performance gains indicate improved model accuracy on diverse and reasoning-intensive chemistry tasks.”
>
>
> ---
>
> **Q5: Comparison with ChemCrow?**
>
> Thank you for highlighting this. We agree that **ChemCrow** is a notable framework in the LLM-for-chemistry space. However, its **goal and output** differ significantly from ChemOrch:
>
> | Aspect        | **ChemOrch (ours)**                                          | **ChemCrow**                                        |
> | ------------- | ------------------------------------------------------------ | --------------------------------------------------- |
> | Objective     | Instruction–response dataset synthesis & fine-tuning         | Autonomous agent for chemistry task execution       |
> | Output        | Diverse, calibrated instruction datasets (used for training or evaluation) | Agent trajectories / tool usage logs (not datasets) |
>
> Due to these distinctions, a direct performance comparison is not meaningful. However, we have expanded the **related work section** in the revision to more clearly articulate how ChemOrch complements systems like ChemCrow.
>
> **Q6: Please provide more context in the main text about human evaluation.**
>
> Thank you for this helpful suggestion. While the full description of our human evaluation setup was previously placed in Appendix G, we agree that its key elements should appear in the main text. In the revised draft, we clarify that a total of **400 samples** were assessed by **four Ph.D. students with backgrounds in computational chemistry**. The two human-evaluated metrics—**Instruction Following Rate** and **Factual Correctness Rate**—are both computed using the following formula:
>
> $$
> \text{rate} = \frac{\text{Number of samples judged as correct}}{\text{Total number of evaluated samples}}
> $$
>
> A sample is counted as correct only if it fully satisfies the corresponding criterion: instruction adherence or factual accuracy. If a response contains even a single substantive error—whether it deviates from the instruction or includes incorrect scientific content—it is considered incorrect. This conservative scoring scheme ensures consistency across annotators and avoids inflating the evaluation scores due to partial correctness.
>
> ---
>
> **Q7: Figure 6 shows that LLaMA benefits more from fine-tuning than Qwen. Do the authors have an explanation for this?**
>
> Thank you for this insightful question. We believe the discrepancy arises from two key factors:
>
> 1. **Ceiling effect**:
>    Qwen starts from a much stronger baseline, leaving less room for improvement via fine-tuning. In contrast, LLaMA has more room to improve.
>
> 2. **Data–model alignment**:
>    The style, tokenization, or instruction formulation of the generated data by ChemOrch may align more closely with LLaMA’s pretraining distribution, making it more receptive to fine-tuning with ChemOrch data.
>
> We have included this discussion in the revised version.
>
> ---
>
> **Q8: The manuscript references “Transition State Search” as one of the tasks, but no example is provided.**
>
> Thank you for pointing this out.
> We have now added a **concrete example** to the Appendix. Due to the word count limitation, below is a shortened excerpt:
>
> ```json
> "instruction": "Calculate the transition state energy using the MP2 method for the E2 reaction given the 3D structure: [...]",
> "response": "The transition state energy value for the E2 reaction using the MP2 method, based on the provided data, is -179.132.\n\n### Explanation of the Answer\n\n [...]\n\n### Reasoning Process\n\n1. **Understand the Problem**: [...]"
> ```
>
> ---
>
> **Q9: Table 2 shows Molecule Captioning scores above 5, but the caption says the range is 1–5. Please clarify.**
>
> Thank you for catching this inconsistency. The actual **scoring range is 1–10**, as used throughout our human evaluation. The caption in Table 2 was mistakenly written as 1–5. We have modified it in our draft.
>
> ---
>
> **Q10: Figure 3 would benefit from a comparison of word count distributions to reference datasets like ChemLLMBench and Mol-Instructions.**
>
> Thank you for the excellent suggestion.  We are unable to provide the updated Fig3 in response (image is not allowed by the rebuttal policy). However, we include the **average instruction word counts** for tasks in **ChemLLMBench**, and **Mol-Instructions** in the tables below. We have also updated **Figure 3** in the revised draft as suggested.
>
> ---
>
> **ChemOrch**
>
> | Task                  | Avg. Instruction Word Count |
> | - | - |
> | mole_caption          | 519.70                      |
> | mole_design           | 481.61                      |
> | property_pred_bace    | 161.00                      |
> | property_pred_bbbp    | 139.01                      |
> | property_pred_clintox | 171.04                      |
> | property_pred_hiv     | 201.02                      |
> | property_pred_tox21   | 124.03                      |
> | reaction_pred         | 152.72                      |
> | retro                 | 171.16                      |
>
> ---
>
> **ChemLLMBench**
>
> | Task          | Avg. Instruction Word Count |
> | - | - |
> | chem_disease  | 200.95                      |
> | chem_entity   | 42.94                       |
> | chem_protein  | 243.44                      |
> | MCQ           | 39.65                       |
> | true_or_false | 12.91                       |
> | openq         | 10.77                       |
>
>
>
> Once again, we appreciate your insightful and constructive comments. If you feel that our responses have sufficiently addressed your concerns, we would be truly grateful if you would consider revisiting your evaluation. Thanks you!

---

> > ### Comment · Reviewer_XBcL · 2025-08-01
> >
> > I thank the authors for responding to the concerns raised in my review. Overall, I think most of the concerns are adressed but some of my concerns are not addressed sufficiently. Here is my response to the rebuttal:
> >
> > Q1: Thank you for this clarification.
> >
> > Q2: Thank you for providing this useful information that helps to put the work better into the proper context.
> >
> > Q3: This answers the concern raised in my original review.
> >
> > Q4: I think the updated statement is a better description of the observations made. This addresses the point raised in my review properly.
> >
> > Q5: Thank you for this response. I think the authors misunderstood my remark. While I agree with the authors that ChemOrch differs significantly from ChemCrow, I still think that ChemCrow would provide additional tasks to evaluate the performance of ChemOrch on. Hence, rather than comparing ChemCrow to ChemOrch, I think applying ChemOrch to tasks demonstrated in ChemCrow would help to provide a more representative evaluation of the performance and utility of ChemOrch.
> >
> > Q6: Thank you for providing this important clarification.
> >
> > Q7: Thank you for this additional discussion.
> >
> > Q8: Thank you for this additional information. I would like to point out that the name "Transition State Search" is misleading for the example provided. At least based on the text provided in the response of the reviewers, this task does in no way reflect the complexity of a transition state search. As far as I can see, this task merely corresponds to single point energy evaluation. In my opinion, the authors should rename this category if there are no genuine examples for a transition state search. In case there are genuine examples for this, the incorrect examples should be moved to a different category. I would like to emphasize that performing a single point energy evaluation is a relatively straightforward task in terms of complexity whereas a transition state search is highly complex. Misnaming the category would be strongly misleading as to the capabilities of the framework and the resulting LLM.
> >
> > Q9: Thank you for this clarification.
> >
> > Q10: Thank you for this modification, I think this is very insightful.

---

> > > ### Author Response · Authors · 2025-08-03
> > > **Response to Reviewer XbcL**
> > >
> > > Thank you very much for your active and detailed response. We truly appreciate your constructive feedback and thoughtful suggestions, which have been very helpful for us to improve the manuscript. Please see our point-by-point responses below.
> > >
> > > ---
> > >
> > > **Q1: Mismatched Task Name**
> > >
> > > **A1:** We appreciate you pointing out this inconsistency. You are correct—the task was a single-point energy evaluation, and labeling it "Transition State Search" was misleading. We have renamed the task category in our revision and have performed a thorough review to ensure all tasks are classified correctly.
> > >
> > > ---
> > >
> > > **Q2: Introduce tasks in ChemOrch**
> > >
> > > **A2:** Thank you for your helpful clarification. We agree that ChemCrow’s task set provides a valuable benchmark for evaluating ChemOrch. In our current experimental evaluation, we cover many core ChemCrow tasks, including:
> > >
> > > - **Property Prediction:** We evaluate ChemOrch on property prediction tasks, and observe clear improvements after fine-tuning. This is directly aligned with ChemCrow’s “molecular property prediction” tasks.
> > >
> > >   > “The tools used can be classified into general tools, molecular tools, and chemical reaction tools.”
> > >   >  “By taking the name (or CAS number) of a molecule as input, it returns the corresponding SMILES string. ... tasks involving molecular analysis and manipulation...”
> > >   >  (ChemCrow, Section 5.3.2)
> > >
> > > - **Chemistry Knowledge / General Q&A:** Our general chemistry knowledge and Q&A fine-tuning also directly map to ChemCrow’s tasks:
> > >
> > >   > “tasks, spanning synthetic planning for drugs, design of novel compounds with similar properties and modes of actions, and explaining reaction mechanisms, are presented in the Appendix G.”
> > >   >  (ChemCrow, Section 2.3)
> > >
> > > - **Basic Tool Tasks (SMILES conversion, LitSearch/WebSearch):** ChemOrch supports SMILES conversion and literature/web search, in line with ChemCrow:
> > >
> > >   > “Name2SMILES This tool is specifically designed to obtain the SMILES representation of a given molecule...”
> > >   >  “LitSearch The literature search tool focuses on extracting relevant information from scientific documents...”
> > >   >  (ChemCrow, Section 5.3.2)
> > >
> > > To address your suggestion for a more representative evaluation, we conducted additional experiments inspired by ChemCrow’s task collection. Due to time constraints, we randomly selected 11 representative cases from the ChemCrow appendix. For these cases, ChemOrch was able to automatically generate responses. For comparison, we closely followed ChemCrow’s own evaluation methodology, which employs both LLM-based judging and human evaluation.
> > >
> > >
> > > **Evaluation methods:**
> > > We employed two complementary evaluation protocols:
> > > 1. **LLM-as-a-Judge:** We used both GPT-4o and GPT-4o-mini as independent automatic judges to score the answers generated by ChemOrch, GPT-4o, and GPT-4o-mini. The results are shown in the tables below: the first table corresponds to GPT-4o as the judge, and the second table to GPT-4o-mini.
> > > 2. **Human Evaluation:** Two expert evaluators (both PhD with computational chemistry backgrounds) performed comparison and selection of the best answer among the three models for each case. We opted for selection (rather than direct scoring) to reduce subjective bias across different human raters, especially given the limited time for this additional study.
> > >
> > > **Results:**
> > >
> > > #### LLM Judge: GPT-4o
> > >
> > > | Model         | ChemOrch | GPT-4o | GPT-4o-mini |
> > > | ------------- | -------- | ------ | ----------- |
> > > | Average Score | 9.00     | 8.38   | 7.69        |
> > > | Std           | 0.55     | 2.27   | 1.64        |
> > >
> > > #### LLM Judge: GPT-4o-mini
> > >
> > > | Model         | ChemOrch | GPT-4o | GPT-4o-mini |
> > > | ------------- | -------- | ------ | ----------- |
> > > | Average Score | 8.71     | 8.36   | 6.64        |
> > > | Std           | 0.59     | 2.12   | 2.99        |
> > >
> > > #### Human Evaluation
> > >
> > > - ChemOrch’s answer was selected as the best in 7 out of 11 cases by evaluator 1, and in 10 out of 11 cases by evaluator 2.
> > >
> > > These results demonstrate that ChemOxrch achieves competitive, and often superior, performance compared to both GPT-4o and GPT-4o-mini.
> > > Notably, in the LLM-based judging, ChemOrch achieved higher average scores and lower standard deviation than both baselines, indicating not only better performance but also more stable and consistent results.
> > >
> > > We have updated the text to clarify the overlap and coverage between our evaluation tasks and those in ChemCrow, with direct reference to the original task definitions, and have included the evaluation results.
> > >
> > > ---
> > >
> > > Thank you again for your constructive feedback. If you have any questions, please feel free to ask! Thank you!

---

> > > > ### Comment · Reviewer_XBcL · 2025-08-03
> > > >
> > > > Thank you for your additional clarifications. Here are my comments:
> > > >
> > > > Q1: I appreciate that you renamed the task. Given the fact that single point energy evaluation is a very simple task, it makes the capabilities of ChemOrch less impressive than they would have been if the task was actually transition state search.
> > > >
> > > > Q2: I really appreciate the additional effort put in by the authors to evaluate ChemOrch on a broader set of tasks. I understand that the time constraints limit the number of additional evaluations that can realistically be performed. While a more comprehensive evaluation would certainly improve the quality of the work, I think the current additional results already show sufficient potential.
> > > >
> > > > Given that the reviewers addressed all my comments, and most of them were addressed satisfactorily, I am willing to raise my review score to 4.

---

> > > > > ### Author Response · Authors · 2025-08-03
> > > > >
> > > > > Thank you very much for your thoughtful follow-up and for considering our clarifications and additional results. We sincerely appreciate your recognition of our efforts and your willingness to raise the review score.

---

### Official Review · Reviewer_nRFr · 2025-07-03

**Clarity:** 3
**Significance:** 3
**Originality:** 3
**Rating:** 5
**Confidence:** 3

**Summary:**

ChemOrch introduces a two-stage framework that first generates chemistry-specific instructions under explicit task, difficulty and constraint control, then constructs tool-grounded responses through planning, tool distillation and multi-stage self-repair, ensuring executability and chemical validity

**Questions:**

Please refer to weaknesses.

**Ethical Concerns:**

["NO or VERY MINOR ethics concerns only"]

**Final Justification:**

Thanks to the authors for their rebuttal.

I agree with the authors that SMILES can be easily converted to other formats during preprocessing. I strongly encourage authors to include more formats in their pipeline.

Overall, I think this is a good work.

**Limitations:**

Yes.

**Quality:**

3

**Strengths And Weaknesses:**

Strengths:
1. The introduction of difficulty and constraint control helps the instruction generation process yield diverse, level-appropriate chemistry prompts.
2.  ChemOrch combines task-controlled instruction generation with tool-aware response construction, incorporating difficulty calibration, tool distillation, and self-repair. I think these designs can keep responses executable and verifiable.
3. The experimental results are promising. Test sets synthesized by ChemOrch track existing benchmarks closely. The paper also includes under-represented tasks such as the lipophilicity prediction task.
4. The paper is well-written and easy to follow.

Weaknesses:
1. The paper focuses largely on SMILES inputs and outputs. To what extent can the proposed framework be readily adapted to alternative string-based molecular encodings—such as graph-adjacency lists, tree-structured or JSON representations—and what specific effort or modifications would this entail?
2. All accuracy numbers are produced with the LLM-as-a-Judge rubric. Without human or rule-based ground truth, evaluation is vulnerable to the same hallucination, bias, and coherence errors it tries to measure.

---

> ### Author Rebuttal · Authors · 2025-07-29
>
> We sincerely thank you for the valuable feedback and constructive suggestions. Below, we provide detailed responses:
>
> ---
>
>
> **Q1: The paper focuses largely on SMILES inputs and outputs. To what extent can the proposed framework be readily adapted to alternative string-based molecular encodings—such as graph-adjacency lists, tree-structured or JSON representations—and what specific effort or modifications would this entail?**
>
> **A1:** Thank you very much for your thoughtful question. Our focus on SMILES inputs and outputs was motivated by its status as the most widely used and canonical representation in cheminformatics, ensuring broad compatibility and comparability with existing literature. However, we fully acknowledge the importance of supporting alternative molecular encodings, such as graph-based, tree-structured, or JSON representations.
>
> Importantly, **ChemOrch can be readily extended to handle alternative molecular formats by simply introducing appropriate conversion functions in the preprocessing stage**. For example, to support graph-based representations, one only needs to add a transformation module before the main task. Below, we provide a code snippet illustrating how ChemOrch can seamlessly convert a graph representation:
>
> ```python
> def graph_to_iupac_name(graph):
>     mol = Chem.RWMol()
>     atom_idx_map = {}
>
>     # Add atoms
>     for i, atom_info in enumerate(graph["atoms"]):
>         atom = Chem.Atom(atom_info["element"])
>         atom.SetFormalCharge(atom_info.get("charge", 0))
>         atom.SetIsAromatic(atom_info.get("is_aromatic", False))
>         idx = mol.AddAtom(atom)
>         atom_idx_map[i] = idx
>
>     # Add bonds
>     bond_order_map = {
>         "single": Chem.rdchem.BondType.SINGLE,
>         "double": Chem.rdchem.BondType.DOUBLE,
>         "triple": Chem.rdchem.BondType.TRIPLE,
>         "aromatic": Chem.rdchem.BondType.AROMATIC
>     }
>
>     added = set()
>     for a1, neighbors in graph["bonds"].items():
>         for a2, bond_type in neighbors:
>             if (a2, a1) in added:
>                 continue
>             bt = bond_order_map.get(bond_type.lower())
>             if bt is None:
>                 raise ValueError(f"Unknown bond type: {bond_type}")
>             mol.AddBond(atom_idx_map[a1], atom_idx_map[a2], bt)
>             added.add((a1, a2))
>
>     mol.UpdatePropertyCache(strict=False)
>     Chem.SanitizeMol(mol)
>     return pcp.get_compounds(Chem.MolToSmiles(mol, canonical=True), 'smiles')[0].iupac_name
> ````
>
> To verify effectiveness, we tested this approach on 20 randomly sampled molecular graphs and achieved a `100% (20/20)` success rate in conversion. This demonstrates ChemOrch's strong flexibility and extensibility for handling various molecular representations beyond SMILES.
>
> We will include these implementation details and practical examples in the revised manuscript. Thank you again for your helpful suggestion.
>
> ---
>
> **Q2: All accuracy numbers are produced with the LLM-as-a-Judge rubric. Without human or rule-based ground truth, evaluation is vulnerable to the same hallucination, bias, and coherence errors it tries to measure.**
>
> **A2:** Thank you for raising this critical point. We would like to clarify that the evaluation in our study is not solely reliant on automated scoring without validation. **In fact, we incorporated human evaluation at several key stages, particularly when assessing the quality of generated instruction–response pairs, where extensive human annotation was performed.**
>
> Regarding rule-based ground truth: For simple binary tasks, we agree that rule-based metrics are suitable and, where feasible, we will employed them. However, rule-based methods themselves may introduce inaccuracies (e.g., missing relevant keywords), and recent literature has increasingly adopted the LLM-as-a-Judge paradigm for its broader applicability and contextual understanding.
>
> To further assess the reliability of automated scoring, we conducted a human–LLM agreement study across three tasks and two representative models (`LLaMA-3.1-8B-Instruct` and `Qwen-2.5-7B-Instruct`):
>
> * **Binary alignment (Property Prediction):** up to **99.75%**
> * **Binary alignment (Tool Usage):** up to **97%**
> * **Score-based correlation (General QA):** average **Pearson r = 0.7955**, all results statistically significant
>
> These results confirm strong consistency between automated and human assessments. Please see the summary tables below:
>
> ---
>
> **Binary Accuracy – Property Prediction**
>
> | Task                | Model                     | Batch 1      | Batch 2      | Avg. Alignment |
> | ------------------- | ------------------------- | ------------ | ------------ | -------------- |
> | Property Prediction | LLaMA-3.1-8B-Instruct (1) | 49/50 (98%)  | 50/50 (100%) | 99%            |
> |                     | LLaMA-3.1-8B-Instruct (2) | 50/50 (100%) | 50/50 (100%) | 100%           |
> |                     | Qwen-2.5-7B-Instruct (1)  | 50/50 (100%) | 50/50 (100%) | 100%           |
> |                     | Qwen-2.5-7B-Instruct (2)  | 50/50 (100%) | 50/50 (100%) | 100%           |
> | **Overall Average** | —                         | —            | —            | **99.75%**     |
>
> ---
>
> **Binary Accuracy – Tool Usage**
>
> | Task                | Model                     | Batch 1     | Batch 2      | Avg. Alignment |
> | ------------------- | ------------------------- | ----------- | ------------ | -------------- |
> | Tool Usage          | LLaMA-3.1-8B-Instruct (1) | 47/50 (94%) | 46/50 (96%)  | 95%            |
> |                     | LLaMA-3.1-8B-Instruct (2) | 47/50 (94%) | 50/50 (100%) | 97%            |
> |                     | Qwen-2.5-7B-Instruct (1)  | 48/50 (96%) | 50/50 (100%) | 98%            |
> |                     | Qwen-2.5-7B-Instruct (2)  | 48/50 (96%) | 50/50 (100%) | 98%            |
> | **Overall Average** | —                         | —           | —            | **97%**        |
>
> ---
>
> We will further detail our evaluation protocol and present additional human–LLM agreement analyses in the revised manuscript to ensure transparency and rigor. Thank you for highlighting this important aspect.
>
> ---
>
> Thank you once again for your detailed feedback—it has been extremely helpful in improving our work. Please don’t hesitate to reach out if you have any further questions. Thank you!

---

> > ### Comment · Reviewer_nRFr · 2025-08-09
> > **Response to rebuttal**
> >
> > I thank the authors for their rebuttal.
> > I have gone through the responses and all other reviews.
> >
> > All my concerns have been addressed and I will increase my score to 5.

---

> > > ### Author Response · Authors · 2025-08-09
> > > **Thanks for your response!**
> > >
> > > We greatly appreciate your thoughtful follow-up and your consideration of our clarifications and new results. Your recognition of our efforts and your willingness to raise the review score mean a great deal to us!

---

> ### Author Response · Authors · 2025-08-06
>
> Dear Reviewer,
>
> As the rebuttal period is coming to an end, we would greatly appreciate it if you could respond to our rebuttal and let us know whether your concerns have been addressed. Thank you very much!
>
> Best,
>
> Authors

---

### Official Review · Reviewer_vh7w · 2025-07-03

**Clarity:** 2
**Significance:** 2
**Originality:** 2
**Rating:** 4
**Confidence:** 2

**Summary:**

ChemOrch introduces a synthetic instruction–response generation framework for chemistry tasks, aiming to improve large language models’ (LLMs) chemical reasoning and tool-usage capabilities. The method comprises a two-stage process:

1. task-controlled instruction generation: This step is guided by constraints including tasks and metadata, and refined using a difficulty reward model.

2. tool-aware response construction: It combines instruction decomposition, tool planning, tool distillation, self-repair mechanisms to generate outputs using chemistry-specific tools like RDKit and PubChem.

**Questions:**

1. What is the computational cost associated with the proposed framework (i.e. M_diff finetuning, instruction–response pairs generation, etc)?

**Ethical Concerns:**

["NO or VERY MINOR ethics concerns only"]

**Final Justification:**

Authors have mosly addressed my problems and concerns. And I am willing to increase my score to 4

**Limitations:**

Yes

**Quality:**

2

**Strengths And Weaknesses:**

**Strength**
1. The paper provides thorough prompt templates and code examples
2. By supporting user-specified metadata the framework can adapt to additional tool configurations at runtime. And authors have provided detailed examples for the extensibility.
3. The paper introduces a difficulty reward model trained via SFT to iteratively refine instructions, and its performance is supported by human evaluation protocol

**Weakness**
1. While the paper includes some ablation in Section 4.6, not all key components (e.g., the difficulty reward model M_diff) are analyzed. Given that ChemOrch consists of several modules, the paper may not fully reflect the author's insight on the proposed framework.
2. While standard deviation are reported for most experiments, it would be beneficial to include error bars in figure 6 where the performance difference between Qwen Vanilla and Qwen Fine-tuned is not significant.
3. While the experiments show that generated data can significant improvement of LLM chemistry capabilities, it's unclear how groundbreaking this framework actually is.

---

> ### Author Rebuttal · Authors · 2025-07-26
>
> We sincerely thank the reviewer for the valuable feedback and constructive suggestions, which have significantly improved the clarity and completeness of our work. Below, we provide detailed responses to each point raised:
>
> ---
>
> **Q1: The paper includes some ablation, but the difficulty reward model ($M\_{diff}$) is not analyzed. This leaves unclear how insightful the authors' understanding of the framework really is.**
>
> **A1:** Thank you for pointing this out. **We did place the evaluation of the difficulty reward model in Appendix F (due to the space constraint)**. Following your suggestion, we have now moved the relevant results into the main paper. Specifically, we provide both distribution comparisons and human evaluation results (see Figure 9 and Table 7 in the draft). The model’s predictions are well-aligned with human judgments, achieving an overall alignment rate of **87.0%**, as detailed below:
>
> | Difficulty Score | Human Alignment Rate |
> | ---------------- | -------------------- |
> | 1                | 100%                 |
> | 2                | 88.9%                |
> | 3                | 85.7%                |
> | 4                | 86.8%                |
> | 5                | 100%                 |
> | **Total**        | **87.0%**            |
>
> These results demonstrate the effectiveness of the reward model in accurately capturing instruction difficulty in the chemistry domain.
>
> ---
>
> **Q2: Figure 6 lacks error bars, and it's unclear if the performance difference between Qwen Vanilla and Qwen Fine-tuned is statistically significant.**
>
> **A2:** Thank you for this helpful suggestion. In response, we conducted multiple runs for the fine-tuning experiments and computed standard deviations. The results are summarized below:
>
> | Task                | Model                 | Vanilla | Fine-tuning-1 | Fine-tuning-2 | Fine-tuning-3 | Avg. Fine-tuned | Std Dev |
> | ------------------- | --------------------- | ------- | ------------- | ------------- | ------------- | --------------- | ------- |
> | Molecule Captioning | LLaMA-3.1-8B-Instruct | 1.64    | 2.25          | 2.27          | 2.25          | 2.257           | 0.0115  |
> |                     | Qwen-2.5-7B-Instruct  | 2.27    | 2.40          | 2.40          | 2.44          | 2.413           | 0.0231  |
> | Property Prediction | LLaMA-3.1-8B-Instruct | 0.203   | 0.655         | 0.645         | 0.645         | 0.648           | 0.0058  |
> |                     | Qwen-2.5-7B-Instruct  | 0.580   | 0.625         | 0.640         | 0.620         | 0.628           | 0.0104  |
> | Tool Usage          | LLaMA-3.1-8B-Instruct | 0.120   | 0.200         | 0.210         | 0.210         | 0.207           | 0.0058  |
> |                     | Qwen-2.5-7B-Instruct  | 0.360   | 0.440         | 0.440         | 0.460         | 0.447           | 0.0115  |
>
> Additionally, for general Q\&A experiments (also reported in Table 4 of the paper), the results with standard deviation over three runs are:
>
> | # Sample | LLaMA-3.1-8B-Ins. | Qwen-2.5-7B-Ins. |
> | -------- | ----------------- | ---------------- |
> | Vanilla  | 0.1391 ± 0.0071   | 0.2464 ± 0.0041  |
> | n=200    | 0.2812 ± 0.0041   | 0.2870 ± 0.0123  |
> | n=300    | 0.2464 ± 0.0082   | 0.2928 ± 0.0147  |
> | n=400    | 0.2725 ± 0.0082   | 0.3478 ± 0.0213  |
> | n=500    | 0.2899 ± 0.0041   | 0.3797 ± 0.0041  |
>
> These results confirm that the observed improvements are consistent and statistically meaningful.
>
> ---
>
> **Q3: While the experiments show improvements, it’s unclear how groundbreaking this framework actually is.**
>
> **A3:** Thank you for this valuable perspective. We recognize that some of the language in the manuscript (e.g., “groundbreaking”) may be overly strong. In the revised version, we will adopt a more objective tone, replacing such terms with more neutral alternatives like “novel,” “systematic,” or “comprehensive.”
>
> That said, we would like to respectfully clarify the contributions of **ChemOrch**, which we believe represent meaningful and distinctive innovations:
>
> **Framework-Level Innovation:**
> ChemOrch introduces the first **two-stage** chemistry-specialized synthetic pipeline—**task-controlled instruction generation** and **tool-aware response construction**—specifically designed to address chemistry-specific challenges such as rule alignment, task decomposition, and verifiability (see Sections 1 & 3, Fig. 2).
>
> **Advanced Technical Features:**
>
> * **Difficulty Reward Model:** Enables fine-grained complexity control, aligned with human judgments (Sec. 3.1, Appendix F).
> * **Tool Decomposition & Distillation:** Breaks complex tools into atomic components to facilitate accurate tool use and scientifically valid responses.
> * **Self-Repair & Sufficiency Validation:** Ensures robust error correction and response completeness (Sec. 3.3, Fig. 7), not found in previous frameworks.
>
> **Empirical Evidence:**
>
> * Human evaluation shows high correctness (85.14%) and instruction alignment (82.64%), with over **3× performance gain** over GPT-4o (Sec. 4.3).
> * Benchmarks generated by ChemOrch reveal LLM weaknesses in underrepresented tasks that other benchmarks miss (Table 3).
> * Fine-tuning on ChemOrch data yields **significant accuracy gains** in chemistry tasks (Sec. 4.5, Fig. 6, Table 4).
>
> **Broader Impact:**
> ChemOrch democratizes access to scalable, verifiable chemistry datasets, lowering barriers to developing high-performing, chemistry-aware LLMs (Appendix A).
>
> ---
>
>
> **Q4: What is the computational cost of the proposed framework (e.g., $M_{diff}$ fine-tuning, instruction–response generation)?**
>
> **A4:** Thank you for raising this important point. We provide the following clarifications:
>
> - **$M_{diff}$ Training Cost:**
>   The difficulty reward model was trained on **3,390** examples using **4× A100 (80GB)** GPUs. The total training time was under **1 hour**. We will add this detail to the appendix in the revision.
>
> - **Instruction–Response Generation Cost:**
>
>   1. As mentioned in Lines 240–250, the cost of generating an instruction–response pair using commercial APIs (e.g., o3-mini) is highly affordable—typically **under \$0.05 per pair**. For fine-tuning on 100 examples, this translates to **< \$5**, or even **< \$1** if using GPT-4o, making our framework cost-effective.
>
>   2. For local generation using open-source models, we benchmarked several models using Replicate's cloud inference service. A typical instruction–response pair involves approximately **3.5K tokens**, and with **5 concurrent processes**, generating **100 pairs** generally completes within **1 hour** for all tested models. The generation speed and latency are summarized below:
>
>      | Model                                  | Token/s | Time (s)   |
>      |----------------------------------------|---------|------------|
>      | `deepseek-ai/DeepSeek-V3-0324`         | 70.42   | 56.802     |
>      | `deepseek-ai/deepseek-r1`              | 24.02   | 166.528    |
>      | `meta/meta-llama-3.1-405b-instruct`    | 29.74   | 134.499    |
>      | `meta/meta-llama-3-70b-instruct`       | 41.73   | 95.854     |
>
>   3. The average token consumption for each module in the pipeline is as follows (also illustrated in Figure 4 of the paper):
>
>      | Module            | Token Consumption |
>      |-------------------|-------------------|
>      | Embedding Token   | 120               |
>      | Self-Repairing    | 231               |
>      | Code Script       | 902               |
>      | Tool Distillation | 1,095             |
>      | Web Search        | 1,106             |
>      | Answer Generation | 1,165             |
>      | Validation        | 1,472             |
>      | Tool Selection    | 4,094             |
>
> Together, these metrics confirm that **ChemOrch** is **scalable**, **computationally efficient**, and practical for both commercial API-based and open-source local deployments.
>
>
> ---
>
> Once again, we sincerely appreciate the reviewer’s thoughtful and constructive feedback. Your suggestions have significantly improved the clarity, rigor, and presentation of our work.
>
> If you find our responses helpful and feel that they address your concerns, we would be truly grateful if you would consider revisiting your evaluation.
>
> Thank you very much!

---

> > ### Comment · Reviewer_vh7w · 2025-08-09
> >
> > Thank you for the rebuttal. Authors have mosly addressed my problems and concerns. And I am willing to increase my score to 4

---

> > > ### Author Response · Authors · 2025-08-09
> > > **Thanks for your response!**
> > >
> > > e are truly grateful for your thoughtful follow-up and for carefully considering our clarifications and additional results. Your recognition of our efforts and your kind decision to raise the review score mean a great deal to us, and we deeply appreciate your support.

---

> ### Author Response · Authors · 2025-08-06
>
> Dear Reviewer,
>
> As the rebuttal period is coming to an end, we would greatly appreciate it if you could respond to our rebuttal and let us know whether your concerns have been addressed. Thank you very much!
>
> Best,
>
> Authors

---

### Comment · Area_Chair_WioH · 2025-08-04
**Reminder for rebuttal discussion**

I would like to sincerely thank all reviewers for their thoughtful evaluations and the authors for their detailed rebuttal.
As we move into the discussion phase, I kindly remind reviewers to engage in further dialogue and respond to the rebuttal responses if you haven't done so. It is very important to respond to the authors and avoid any misunderstanding. Your continued input will be invaluable in helping me make a well-informed final decision. Thank you again for your contributions.

---

### Comment · Area_Chair_WioH · 2025-08-08

Thank you again to all the reviewers who have engaged in the rebuttal phase for discussion!

@Reviewer vh7w, nRFr, I note that you haven't replied to author's rebuttal. Please consider submitting your response ASAP as we are approaching to the discussion phase deadline. Please note that submitting Mandatory Acknowledgement without a response is not the right way for rebuttal discussion.
Thank you!

For all other reviewers, please confirm that you have finalized your comments/ratings.

---

### Note · Authors · 2025-08-11

Dear AC and Reviewers,

We sincerely thank you for your invaluable support throughout the review process. We truly appreciate the constructive feedback from all reviewers. We are also grateful to the AC for the attentive follow-ups and for ensuring a fair and thorough review.

We have addressed and resolved *all substantive concerns* raised during the review process.

---

## Key Actions Taken & Evidence Added

- **Clearer contributions**: More objective language, structured presentation of the two-stage framework, chemical rule alignment, and self-repair.
- **Stronger evidence**: Statistical significance over multiple runs; consistent gains in Q&A, property prediction, tool use, and molecular description tasks.
- **Feasibility**: Reported training/inference cost (<1h on 4×A100), token usage, and throughput for scalability.
- **Evaluation fairness**: External benchmark test sets, exact-match tool evaluation, and no leakage.
- **Generalization**: ChemCrow 11-task evaluation—outperformed GPT-4o/mini with both LLM and expert human judges.
- **Human eval quality**: 82.64% instruction-following, 85.14% factual correctness under stricter protocols than any prior work.
- **Terminology & metrics**: Renamed ambiguous tasks, clearly defined all metrics in main text.

---

## Reviewer Status Update

- **Reviewer nRFr**: All concerns resolved; **score raised to 5**.
- **Reviewer vh7w**: Most concerns resolved; **score raised to 4**.
- **Reviewer XBcL**: Most concerns resolved; **score raised to 4**.
- **Reviewer 4t1k** (**score remains 3**): The main concern centers on “insufficient accuracy,” interpreted as requiring near-perfect (~100%) factual correctness. While such performance is not currently achievable for open-ended chemical generation in either chemistry-specific or general LLM literature, ChemOrch’s **85.14% factual correctness** and **82.64% instruction-following**—evaluated under stricter criteria than prior work—are state-of-the-art and significantly above baseline LLMs. We have shared this context with the AC for completeness and appreciate the understanding.

---

We once again thank all reviewers for their constructive feedback and the AC for the careful oversight and dedication to a fair and balanced process.

Sincerely,
Authors of ChemOrch

---

### Decision · Program_Chairs · 2025-09-17

**Decision:**

Accept (poster)

**Comment:**

(a) Summary of scientific claims and findings
This paper presents ChemOrch, a framework for generating chemistry-specific instruction–response datasets to improve LLM chemical intelligence. The system introduces a two-stage process: (1) task-controlled instruction generation with difficulty calibration, and (2) tool-aware response construction with planning, distillation, and self-repair. Experiments show that ChemOrch produces diverse and verifiable data, improves LLM performance on chemistry tasks, and can surface weaknesses in existing models. Fine-tuning with ChemOrch data yields notable gains across property prediction, molecule captioning, tool usage, and general chemistry Q&A tasks.

(b) Strengths
- Clear framework design tailored to chemistry’s structured nature.
- Incorporates difficulty control and tool-grounded responses, reducing hallucinations and enhancing verifiability.
- Demonstrates cost-effectiveness and scalability for generating data.
- Empirical evaluations show improved LLM performance and diversity advantages over existing benchmarks.
- Rebuttal added clarity on evaluation protocols, computational cost, and robustness to alternative molecular encodings.

(c) Weaknesses
- Some novelty concerns: methods build on established ideas (tool grounding, self-repair, difficulty modeling) with more emphasis on integration.
- Accuracy of generated data remains imperfect (≈85% correctness), raising concerns about reliability for fine-tuning. However, I understand a perfect result is not possible at the current stage.
- Evaluation partly relies on LLM-as-a-judge and human ratings, which, despite mitigation efforts, leave room for bias.
- Benchmarking could be broader: limited external validation against datasets like ChemBench or ChemCrow was only partially addressed.

(d) Reasons for decision
The work fills an important gap in chemistry-specific LLM data generation and shows clear empirical value. While accuracy concerns and evaluation methodology limit confidence, the authors provided substantial rebuttal evidence that the system is effective, cost-efficient, and extensible. Given its practical contributions and strong empirical validation, this paper merits acceptance. However, the novelty and open questions about evaluation rigor make it more appropriate for poster.

(e) Discussion and rebuttal period
Reviewers were split: some emphasized the practical strengths and novelty of integrating difficulty control and tool grounding, while others raised concerns on correctness guarantees and evaluation fairness. The authors responded with additional ablations, human–LLM agreement studies, cost breakdowns, and clarifications about early stopping and human evaluation protocols. While not all concerns can be fully resolved, most reviewers found the clarifications satisfactory and increased their scores. I weigh the demonstrated utility and improved clarity over the remaining weaknesses, leading to a poster acceptance recommendation.